Brief Communication

# Structure of the human 20S U5 snRNP

Sarah Schneider [1,6], Irina Brandina [1,6], Daniel Peter[1,2,6], Sonal Lagad [1], Angelique Fraudeau[1], Júlia Portell-Montserrat [1,3,4], Jonas Tholen [1,5], Jiangfeng Zhao[1] & Wojciech P. Galej [1]✉

The 20S U5 small nuclear ribonucleoprotein particle (snRNP) is a 17-subunit RNA–protein complex and a precursor of the U4/U6.U5 tri-snRNP, the major building block of the precatalytic spliceosome. CD2BP2 is a hallmark protein of the 20S U5 snRNP, absent from the mature tri-snRNP. Here we report a high-resolution cryogenic electron microscopy structure of the 20S U5 snRNP, shedding light on the mutually exclusive interfaces utilized during tri-snRNP assembly and the role of the CD2BP2 in facilitating this process.

In eukaryotes, the removal of noncoding introns from pre-mRNAs is catalyzed by the large and dynamic spliceosome complex[1]. The spliceosome assembles de novo on each intron from small nuclear ribonucleoprotein particles (snRNPs) and numerous protein factors. The 39-subunit U4/U6.U5 tri-snRNP is the largest preassembled building block of the spliceosome, which joins the splicing pathway at the precatalytic (pre-B) stage and delivers two components of the RNA catalytic core, the U5 and U6 snRNAs, in their inactive configurations requiring further remodeling[2,3]. Despite recent progress in the mechanistic understanding of spliceosome assembly, the biogenesis and recycling of its building blocks remain elusive.

Several factors associate with the tri-snRNP components but are absent in mature particles; hence, they are believed to play roles in tri-snRNP biogenesis and/or recycling. These include the U5 snRNP binding proteins: AAR2 (refs. 4,5), CD2BP2 (U5-52K; *Saccharomyces cerevisiae* Lin1)[6,7], TSSC4 (refs. 8,9) and ZNHIT2 (refs. 10–12), as well as the U4/U6 annealing factor SART3 (*S. cerevisiae* Prp24)[13–15]. Mechanistically, the exact roles and the interplay of these assembly factors remain poorly understood.

The 20S U5 snRNP isolated from HeLa cells contains at least 16 subunits, including its hallmark protein CD2BP2 (refs. 16,17), which also plays a role in the binding of the CD2 receptor[18]. Conditional knockout (KO) of CD2BP2 in mice leads to growth defects and premature death during embryonic development[19]. In its role as a splicing factor, CD2BP2 binds to DIM1 (U5-15K) with its GYF domain, forming a protein–protein interface that differs from the canonical binding mode to sequence motifs in the CD2 antigens[16,18]. Lin1, the yeast homolog of CD2BP2, was reported to

bind PRP8, suggesting a possible mode of its recruitment to the U5 snRNP complex[20].

Although 20S U5 snRNP was first isolated several decades ago[17], its molecular structure and the function of CD2BP2 remain unknown. In this Brief Communication, we investigate how CD2BP2 interacts with other components of the U5 snRNP and how it facilitates tri-snRNP formation.

First, we analyzed the steady-state composition of the spliceosomal snRNPs in the absence of CD2BP2. We purified snRNP using anti-2,2,7-trimethylguanosine (TMG) antibody-coupled resin from nuclear extracts (NE) prepared either from wild-type (WT) HEK293T cells or a homozygous CRISPR–Cas9 CD2BP2 KO cell line (*CD2BP2*[KO]; Extended Data Fig. 1a–c). The composition of both samples was compared by quantitative mass spectrometry (Fig. 1a). As expected, we observe a clear depletion of the CD2BP2 in the KO sample as well as a subtle, but consistent, underrepresentation of nearly all U5 snRNP subunits in the KO condition. U4/U6 and tri-snRNP specific components are also affected, yet, to a lesser extent, while the U2 snRNP proteins remained virtually unchanged, as their assembly into snRNPs is not expected to depend on CD2BP2. As such, these results point to a subtle defect in the U5 and U4/U6.U5 tri-snRNP assembly in the absence of CD2BP2. Yet, we could not observe a noteworthy impact on the cell viability and in vitro splicing efficiency under the conditions tested (Extended Data Fig. 1e). Interestingly, another U5 snRNP assembly factor, AAR2, is significantly enriched in snRNPs isolated from the *CD2BP2*[KO] cells. This could be due to the upregulation of the AAR2 in the absence of CD2BP2 or due to a failure in the CD2BP2-dependent conversion of AAR2-containing U5 snRNP into 20S U5 snRNP during the biogenesis. The latter seems more probable, as the expression levels of AAR2 in NE of WT and KO cell lines are largely unchanged (Extended

[1]European Molecular Biology Laboratory, EMBL Grenoble, Grenoble, France. [2]Present address: Boehringer Ingelheim RCV GmbH & Co KG, Vienna, Austria. [3]Present address: Institute of Molecular Biotechnology of the Austrian Academy of Sciences, Vienna BioCenter, Vienna, Austria. [4]Present address: Research Institute of Molecular Pathology, Vienna BioCenter, Vienna, Austria. [5]Present address: Department of Structural Biology, Genentech Inc., South San Francisco, CA, USA. [6]These authors contributed equally: Sarah Schneider, Irina Brandina, Daniel Peter. ✉e-mail: wgalej@embl.fr

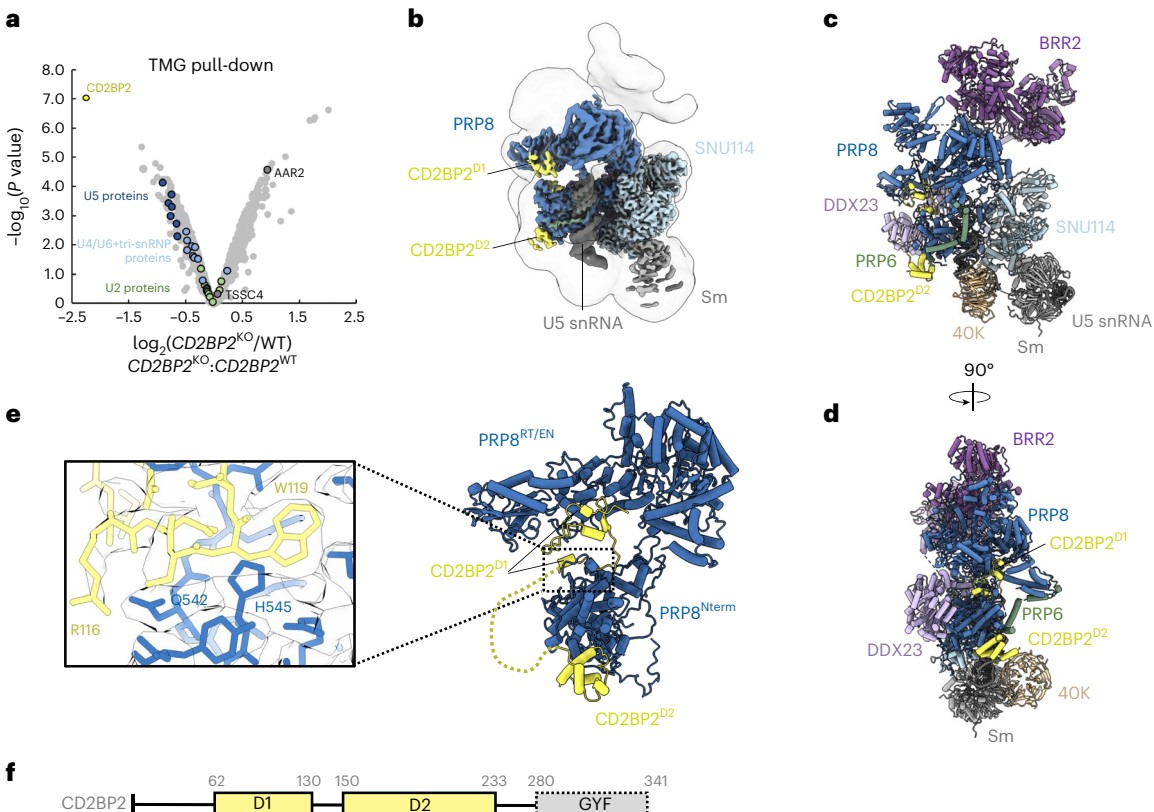

**Fig. 1 | Cryo-EM structure of the 20S U5 snRNP. a**, Quantitative mass spectrometry analysis of snRNPs isolated via TMG agarose from WT or CD2BP2-KO HEK293T cells. A moderated two-sided $t$-test was applied for statistical analysis. **b**, Experimental cryo-EM map of the 20S U5 snRNP colored by the subunit identity fitted into the low-pass filtered map at the lower contour level. **c**, Atomic model of the 20S U5 snRNP shown in the same orientation as in **b**. **d**, Orthogonal view of the atomic model. **e**, Zoomed-in view of the PRP8–CD2BP2 interaction highlighting the extended hook-like domain. **f**, Domain architecture of CD2BP2.

Data Fig. 1g). This data provide evidence that CD2BP2 is indeed involved in the U5 snRNP assembly and establish a functional link to another U5 snRNP assembly factor AAR2.

Next, to gain insights into the structure of the 20S U5 snRNP, we engineered HEK293F cells to express a 3xFLAG_TEV_SBP-tagged CD2BP2 and used it to purify a 17-subunit complex containing the U5 snRNA, seven Sm core proteins and nine other factors (Extended Data Table 1 and Extended Data Fig. 2). The composition of the complex is in good agreement with previous reports[7,17] and, interestingly, includes an additional assembly factor, TSCC4 (not resolved in the structure)[8,9]. We determined a cryogenic electron microscopy (cryo-EM) structure of the CD2BP2-bound 20S U5 snRNP complex at the 3.1 Å resolution (Fig. 1, Table 1, Methods and Extended Data Figs. 3–5). The architecture of the 20S U5 snRNP closely resembles that of the U5 snRNP captured as a part of the tri-snRNP[2,21] and in low-resolution U5 snRNP studies[22]. At least three major states are present in our 20S U5 snRNP reconstruction (Extended Data Fig. 6). State I contains most of the components and is referred to as the 20S U5 snRNP hereafter. State II is missing two helicases, BRR2 and DDX23, and may represent an earlier stage of the U5 snRNP assembly (Extended Data Fig. 6). State III contains particles missing the Sm ring, most probably damaged during the vitrification process. In all reconstructions, PRP8 provides the scaffold for the entire complex and interacts with multiple other subunits (Fig. 1). PRP6 is present in the sample, but only its N-terminal helices are visible, and the tetratricopeptide (TPR) repeat remains disordered. We observed two well-defined densities located near PRP8[RT/En] and PRP8[Nterm] domains, which were assigned to CD2BP2 domains D1 (62-130) and D2 (150-233), respectively (Fig. 1). The CD2BP2[GYF] domain (280-341) and its binding partner

DIM1 are not visible in our structure. A hook-shaped extension of the CD2BP2[D1] bridges the PRP8[RT/En] and PRP8[Nterm] domains and probably stabilizes their relative orientation, which differs from the one observed in the tri-snRNP (Fig. 1 and Extended Data Fig. 7a,b). CD2BP2[D1] occupies the surface of PRP8 that accommodates several different factors during the splicing cycle, including AAR2 in the U5 snRNP precursor[5,23], DIM1 in tri-snRNP and the precatalytic spliceosome[2,21], as well as RNF113 in Bact[24] and CWF19L2 in the postsplicing ILS complexes[25] (Fig. 2 and Extended Data Fig. 7c).

Interestingly, the CD2BP2[GYF] domain delivers DIM1 to the U5 snRNP, which then competes with CD2BP2[D1] for the same binding site on PRP8. The interface of DIM1 that contacts PRP8 in tri-snRNP is most probably occupied by the CD2BP2[GYF] domain, as shown in the crystal structure[16]. Therefore, CD2BP2 constitutes a two-layered buffer blocking the DIM1–PRP8 interaction by simultaneously binding to the interfaces on both PRP8 and DIM1 (Fig. 2). Based on our structural data, we believe that at least two functions of CD2BP2 should be considered. First, it facilitates the recruitment of PRP6 and DIM1, both of which are critical for the tri-snRNP formation. PRP6, together with PRP8, forms an interface that is necessary for PRP31 (and U4/U6 di-snRNP) anchoring, which is then further stabilized by the PRP6[TPR] domain, forming a bridge between U4/U6 and U5 snRNPs (Fig. 2). PRP6 and CD2BP2 interact directly[7] (Extended Data Figs. 8 and 9). Therefore, CD2BP2-mediated prerecruitment of PRP6 to the U5 snRNP would probably enhance the efficiency of the tri-snRNP formation. Second, CD2BP2 acts as a placeholder preventing PRP8[RT/En] association with its numerous binding partners in a wrong spatiotemporal context. A similar mechanism is utilized by some factors involved in ribosome biogenesis[26]. Both of the above functions could serve to ensure the

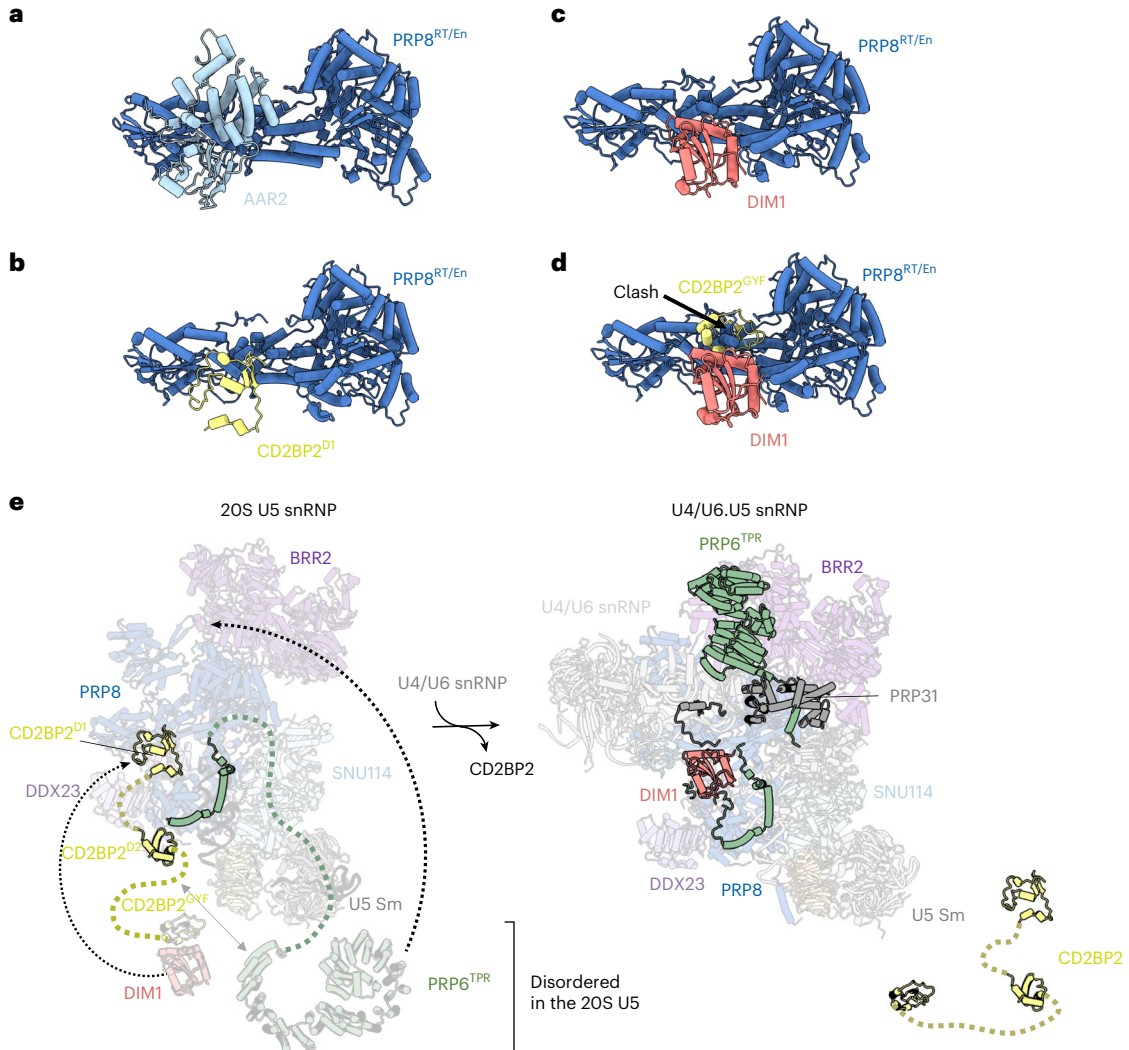

**Fig. 2 | Mutually exclusive interactions of assembly factors with PRP8 during tri-snRNP formation. a**, AAR2 binding mode to PRP8$^{RT/EN}$ domain[23]. **b**, CD2BP2$^{D1}$ occupies an overlapping surface of PRP8 in 20S U5 snRNP. **c**, DIM1 binding site in tri-snRNP[21] is mutually exclusive with CD2BP2$^{D1}$. **d**, PRP8 binding surface on DIM1 is occupied by the CD2BP2$^{GYF}$ domain in the CD2BP2–DIM1 binary complex[16] and clashes with PRP8 when superimposed on DIM1 in tri-snRNP. **e**, A structural model of the tri-snRNP assembly. Left: an atomic model of the 20S U5 snRNP including the disordered DIM1, CD2BP2$^{GYF}$ and PRP6$^{TPR}$ domains. Right: an atomic model of the fully assembled tri-snRNP[21]. Recruitment of the U4/U6 di-snRNP to 20S U5 snRNP triggers the relocation of PRP6, allowing DIM1 to compete with CD2BP2 and displace it from its binding site on PRP8.

correct order of events during the formation of complex intersubunit interfaces within tri-snRNP.

The remaining question is how CD2BP2 displacement is regulated. Our data shows that DIM1 competes with CD2BP2 for the same binding site on PRP8. We could not locate DIM1 in our map, but we observed some additional, low-resolution density near U5 snRNA, which most probably belongs to DIM1/CD2BP2$^{GYF}$ (Extended Data Fig. 6 and 9). Our cross-linking mass spectrometry (XL-MS) data detect DIM1–CD2BP2$^{GYF}$ and 40K–CD2BP2$^{GYF}$ interactions consistent with this putative location (Extended Data Fig. 9). Therefore, it is possible that DIM1 remains constrained in this position and cannot engage in the competition with CD2BP2$^{D1}$. Recruitment of the U4/U6 di-snRNP would trigger a large-scale movement of PRP6 (Fig. 2). Since PRP6 and CD2BP2 interact with one another (Extended Data Figs. 8 and 9), such movement of PRP6 could exert a force on CD2BP2, displacing it from PRP8 and liberating DIM1, allowing it to adopt its final location.

It has been previously shown that CD2BP2 undergoes phosphorylation, which in principle, could also regulate its displacement[27,28].

One of the putative phosphorylation sites lies at the interface between CD2BP2$^{D1}$ and PRP8$^{Nterm}$ (Extended Data Fig. 8) and could potentially modulate their affinity. However, phosphorylation of CD2BP2 does not appear necessary for the in vitro reconstitution of the tri-snRNP, and its function remains unclear[27].

Although our data indicate that CD2BP2 is required for the tri-snRNP formation and acts downstream of AAR2, we cannot discriminate whether its role concerns predominantly the initial biogenesis of the U5 snRNP or its potential recycling from postsplicing complexes. As such, more work is required to shed light on the interplay between these two factors and their roles in respective pathways.

## Online content

**Table 1 | Cryo-EM data collection, refinement and validation statistics**

| | 20S U5 snRNP (complete) (EMD-19041), (PDB 8RCO) | 20S U5 snRNP (core)(EMD-18267), (PDB 8Q91) |
|---|---|---|
| **Data collection and processing** | | |
| Magnification | 130,000 | |
| Voltage (kV) | 300 | |
| Electron exposure (e⁻ Å⁻²) | 40.5 | |
| Exposure rate (e⁻ per pixel per second) | 4.57 | |
| Defocus range (μm) | 1.5–3.5 | |
| Pixel size (Å) | 1.045 | |
| Symmetry imposed | C1 | |
| Movies collected (no.) | 8506 | |
| Initial particle images (no.) | 490,503 | |
| Final particle images (no.) | 76,918 | |
| Map resolution (Å) | 3.1 | |
| FSC threshold | 0.143 | |
| Map resolution range (Å) | 2.7–15 | |
| **Refinement** | | |
| Initial model used (PDB code) | 6qw6 | 6qw6 |
| Model resolution (Å) | 3.2 | 3.2 |
| FSC threshold | 0.5 | 0.5 |
| Model resolution range (Å) | 2.7–15 | 2.7–5 |
| Map sharpening B-factor (Å²) | −55 | −55 |
| Model composition | | |
| Nonhydrogen atoms | 42,341 | 28,634 |
| Protein residues | 6,394 | 3,316 |
| Ligands | 1 | 1 |
| B-factors (Å²) | | |
| Protein | 157.6 | 54.4 |
| RNA | 243.6 | 87.2 |
| Root mean square deviations | | |
| Bond lengths (Å) | 0.007 | 0.005 |
| Bond angles (°) | 0.796 | 0.870 |
| Validation | | |
| MolProbity score | 2.16 | 2.21 |
| Clashscore | 11.6 | 14.3 |
| Poor rotamers (%) | 3.1 | 2.2 |
| Ramachandran plot | | |
| Favored (%) | 96.68 | 95.90 |
| Allowed (%) | 3.26 | 4.04 |
| Disallowed (%) | 0.06 | 0.06 |

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

## Methods

### CD2BP2 KO cell line generation

Two guide RNAs were designed to delete 420 bp of the genomic locus near the translation start site of the CD2BP2 gene and cloned into the PX458 vector (pSpCas9(BB)-2A-GFP; a gift from Feng Zhang; Addgene plasmid no. 48138) using pairs of annealed oligonucleotides as follows:

SG1_FW:CACCGaaagtgaccttccaaggcgt+SG1_REV:aaacacg ccttggaaggtcactttC; SG2_FW:CACCGACACTCTTTGGATAGCGATG +SG2_REV:aaacCATCGCTATCCAAAGAGTGTC.

HEK293T cells (ATCC CRL-3216) were seeded in a 6-well plate at a density of $0.3 \times 10^6$ cells per well and incubated for 24 h in Dulbecco's modified Eagle medium (Gibco) supplemented with 5% fetal bovine serum and penicillin/streptomycin (Thermo Fisher Scientific). Then, 1 µg of each guide RNA-containing PX458 plasmid was transfected into the cells using Lipofectamine 2000 (Thermo Fisher Scientific), following the manufacturer recommendations. After 5 days of growth, cells were trypsinized with trypsin-EDTA 0.05% (Thermo Fisher Scientific), and single cells showing green fluorescent protein (GFP) signal were sorted into 96-well plates using a BD FACSAria IIu (BD Biosciences) sorter. Clonal cell lines were expanded over the period of 2 weeks and analyzed for the presence of the desired deletion using polymerase chain reaction (PCR; 52K_FW: GATCCAGAGGGTCCGCTCC; 52K_REV: CCTTCCTCCATCTCCTCCTGC) and western blotting with anti-CD2BP2 antibodies (Thermo Fisher Scientific; PA5-59603; RRID:AB_2639539).

### TMG agarose immunoaffinity chromatography and quantitative mass spectrometry

NEs from HEK293T WT and *CD2BP2*[KO] were prepared following the original Dignam protocol[29]. TMG agarose beads (TMG mouse antibodies, K121, agarose conjugate, Merck NA02A) were preblocked by incubation with 0.1% BSA in phosphate-buffered saline (PBS) buffer for 1 h at 8 °C, then washed with two bead volumes (CV) of the immunoprecipitation (IP) buffer (20 mM Tris-HCl pH 7.9; 150 mM KCl) supplemented with protease inhibitor cocktail (Roche cOmplete). TMG-beads were added to NEs to the final volume of 10% and incubated ON at 8 °C, with shaking. TMG-beads were collected in Mini Bio-Spin chromatography columns, Bio-Rad, by centrifugation for 30 s at 2,500g at 8 °C. After two wash steps with the IP buffer, beads were eluted by boiling for 10 min with the buffer containing 50 mM Tris-HCl pH 7.9, 150 mM KCl and 0.5% sodium dodecyl sulfate (SDS). The eluates from the experiment performed in triplicates were analyzed by a TMT-plex quantitative mass spectrometry as previously described[30].

### Western Blotting

For the TMG pull-down eluates and NEs, equal amounts of total protein or fractions after glycerol gradient were separated by sodium dodecyl sulfate polyacrylamide gel electrophoresis using WedgeWell 4–20% Tris-glycine system, Invitrogen. The transfer to the polyvinylidene difluoride (PVDF) membrane was done in the Trans-Blot Turbo system, Bio-Rad, using a Turbo-transfer buffer. The following primary rabbit polyclonal antibodies were used: CD2BP2 (Sigma, HPA061309), DIM1 (Proteintech, 27646-1-AP), PRP6 (Invitrogen, PA5-61428) and SNU114/EFTUD2 (Invitrogen, PA5-96559). The secondary antibody was goat anti-rabbit IgG HRP-conjugate (Abcam, ab205718). Mouse monoclonal conjugated antibodies were anti-FLAG M2-peroxidase (Sigma, A8592), anti-GAPDH (Invitrogen MA515738HRP) and anti-HA-Tag F-7 HRP-conjugate (SantaCruz, sc-7392). The blots were visualized using Pierce ECL Western Blotting Substrate, Thermo Fisher Scientific, and documented on the ChemiDoc MP imaging system and ImageLab, Bio-Rad.

For the PRP6-CD2BP2 pull-down experiment, HEK293T cells were seeded into 6-well plates 24 h before transfection at a density of 500,000 cells per well in 1.5 ml Dulbecco's modified Eagle medium medium with 10% fetal bovine serum. Plasmids containing 3xHA_ PRP6[270-941] and 3xFLAG_CD2BP2, both under CMV promoters, were mixed 1:1 and a total of 1 µg of DNA was diluted into in 50 µl of opti-MEM

and mixed with 3 µg of polyethylenimine (PEI) MAX 40K in 50 µl of opti-MEM and incubated at room temperature for 10 min. Transfection solutions were added drop by drop to each well. The cells were collected by centrifugation 48 h after transfection, lysed in 400 µl of lysis buffer (150 mM KCl, 20 mM K-HEPES pH 7.8 and 0.1% Triton X-100) and sonicated for 10 s at 30% amplitude. The lysates were cleared by centrifugation in a table-top centrifuge at 20,000g at 4 °C for 30 min. The supernatant was incubated for 2 h with 5% (v/v) of FLAG-agarose to capture the bait protein. Affinity resin was washed three times with ten resin volumes of buffer 3 (150 mM KCl and 20 mM K-HEPES pH 7.8) and subsequently resuspended in SDS sample buffer and heated up to 95 °C for 5 min to release bound proteins. Input and elution fractions were analyzed by western blotting.

A PVDF membrane (Merck) was activated for 5 min in 100% EtOH and incubated for 5 min in the transfer buffer (1× Tris-glycine, 20% EtOH). A wet transfer was performed for 60–90 min at 30 V in an Invitrogen XCell II Blot Module. The membrane was blocked with 5% milk in PBS supplemented with 0.2% Tween 20 (PBST) for 1 h at room temperature. Primary antibodies were added in the following dilutions: anti-HA 1:5,000 ((HA-7) HRP ab49969, Abcam); anti-FLAG 1:5,000 (HRP sigma A8592-.2MG). The membrane was washed three times for 5 min with 20 ml of PBST, and chemiluminescence was detected with an HRP substrate kit (Pierce ECL Western Blotting Substrate) in a ChemiDoc imager (Bio-Rad).

### In vitro splicing assay

AdML-M3 pre-mRNA substrate was obtained by run-off in vitro T7-transcription[31], capped by VCE, NEB and labeled with fluorescein-5-thiosemicarbazide at the 3′ end as previously described[32]. NEs prepared from WT or *CD2BP2*[KO] cells were used. The typical reaction contained 30 mM KCl, 3 mM $MgCl_2$, 2 mM ATP, 20 mM creatine phosphate, 20 nM RNA AdML_M3 RNA substrate and 40% NE. Splicing reactions were assembled in 20 µl volume and incubated for 2 h at 30 °C. RNA was then isolated by phenol/chloroform extraction and ethanol precipitation and analyzed by denaturing 6% polyacrylamide gel electrophoresis in 7 M urea. Fluorescence of the RNA substrate and splicing product was visualized on ChemiDoc MP.

### 3xFLAG_TEV_SBP_CD2BP2 cell line generation

Open Reading Frame of CD2BP2 was cloned into a modified pFLAG_ CMV10 vector containing an N-terminal 3xFLAG_TEV_SBP affinity tag. FreeStyle 293-F cells were transfected with this plasmid, and a stable, polyclonal cell line was derived through G418 antibiotic selection. Expression of the target protein was confirmed by western blot analysis.

### Purification of the 20S U5 snRNP for cryo-EM analysis

Suspension culture of FreeStyle 293-F cells was grown in the FreeStyle medium (Thermo Fisher Scientific) to the density of ~2 × 10⁶ cells ml⁻¹ in an orbital shaker (Infors) at 37 °C, 8% $CO_2$ and 90 rpm. The cell culture was collected by centrifugation, and NE was prepared following the original Dignam protocol[29]. After the final dialysis step, samples were aliquoted and flash-frozen in liquid nitrogen. For each preparation, an aliquot of NE was thawed on ice, and the salt concentration was adjusted to the final 500 mM KCl. EZview Red anti-FLAG M2 Affinity Gel (Sigma) was added to 10% (v/v) of the reaction volume and incubated overnight at 8 °C with shaking. The resin was washed three times with 10 column volumes (CV) of the wash buffer 1 (20 mM K-HEPES pH 7.9, 500 mM KCl, 2 mM $MgCl_2$, 0.1% Igepal CA-630 and 5% glycerol), and complexes were eluted by incubation in 1 CV of wash buffer 1 supplemented with 10% (v/v) of TEV protease (1 mg ml⁻¹), for 3 h at room temperature. A second-step purification was performed by incubation FLAG eluate with 5% total volume of Pierce high-capacity streptavidin agarose for 3 h at 8 °C with shaking. Beads were washed three times with 10 CV of the wash buffer 2 (20 mM K-HEPES pH 7.9, 500 mM KCl, 2 mM $MgCl_2$). Samples were eluted from the resin by several incubations with 0.5 CV

of wash buffer 2 supplemented with 10 mM biotin for 15 min on ice. The eluate was loaded onto a 4 ml 10–30% glycerol gradient containing 20 mM K-HEPES pH 7.9, 500 mM KCl, 2 mM MgCl$_2$, 0.1% Igepal CA-630 and 0–0.1% glutaraldehyde[33] and centrifuged for 16 h at 35,000 rpm at 4 °C (Beckman Coulter Ultracentrifuge Optima L-90K). The peak fraction of the glycerol gradient was analyzed by negative staining EM, and the fractions containing most homogeneous particles were dialyzed against a buffer containing 20 mM K-HEPES pH 7.9, 150 mM KCl and 2 mM MgCl$_2$ and used directly for grid preparation without further manipulations.

## Cryo-EM data collection and processing
The sample was applied to 300 mesh Quantifoil R 1.2/1.3 grids covered with 3 nm continuous carbon, which had been glow-discharged for 30 s at 15 mA at 0.4 mbar using the Pelco EasiGlow. The grids were plunge frozen in liquid ethane after applying 2 µl at 4 °C, 100 % humidity and blotting for 2 s at blot force −5 in a Vitrobot Mark IV. The grids were screened on a Glacios 200 kV microscope equipped with a Falcon III detector and transferred to a Titan Krios microscope operating at 300 kV equipped with a Gatan Energy filter[34]. A total of 8,506 micrographs were recorded using SerialEM[35] and a K2 direct electron detector at a magnification of 130,000×, a defocus between −1.5 and −3.5 µm with a dose rate of 4.6 e$^-$ per pixel per second and inserted energy slit at 20 eV, as well as the 70 µm objective aperture. The total dose was 40.5 e$^-$ Å$^{-2}$, accumulated in 40 frames at a final pixel size of 1.045 Å. All image processing was done using cryoSPARC v3.3 (ref. 36). For preprocessing, we used patch motion correction and determined the contrast transfer function (CTF) parameters using patch CTF estimation. Using the blob picker functionality, 503,581 particles were picked and extracted in a 504-pixel box. After binning two times, the particles were subjected to two-dimensional classification to create templates for template picking, which resulted in 490,503 picked particles. These particles were subjected to two-dimensional classification, ab initio reconstruction, followed by three-dimensional structure heterogeneous refinement until a homogeneous subpopulation of 76,918 particles was identified. Nonuniform refinement resulted in a final 3.1 Å resolution map based on the 0.143 Fourier Shell Correlation (FSC) criterion[37,38]. The obtained map was sharpened by applying a B-factor of −55 Å$^2$.

## Model building and structural analysis
Atomic coordinates of the U5 snRNP components extracted from the structure of the human tri-snRNP[21] (PDB ID: 6qw6) were used as templates for modeling. Individual chains were fitted into the cryo-EM density as rigid bodies using UCSF Chimera[39], the components with well-resolved density were manually adjusted and rebuilt in Coot v0.9.8.5 (ref. 40). Other components with poorly resolved densities (that is, BRR2, PRP8$^{RNaseH}$, PRP8$^{Jab1/MPN}$, DDX23, Sm ring, 40K) were docked into the map as rigid bodies and left in their original form. CD2BP2 binding sites were initially identified by an exhaustive in silico AlphaFold2-based search[41,42] for all possible interactions with other U5 snRNP components, using a previously described approach[43].

Atomic models were initially refined with Refmac Servalcat v5.8.0267 (ref. 44) with secondary structure restraints generated with ProSMART[45] via the CCP-EM software suite[46]. Final models were refined in real space in Phenix[47] and validated in Molprobity[48]. Structural representations for figures were prepared with Pymol (Schrödinger) and ChimeraX[49].

## Cross-linking and mass spectrometry analysis
CD2BP2 complex at 3 mg ml$^{-1}$ was incubated with 0.25 mM or 1 mM BS3 for 30 min at 30 °C with shaking at 600 rpm (ThermoMixer, Eppendorf), and the cross-linking reaction was quenched by the addition of Tris-Cl pH 7.5 to the final concentration of 50 mM and incubated for 10 min at 35 °C at 600 rpm. Then, samples were mixed with 0.05 (v/v) of RapiGest and, after the addition of 10 mM DTT, incubated at 50 °C for 30 min, with shaking at 600 rpm. Subsequently, 2-chloroacetamide

was added to 50 mM final concentration, and samples were incubated at 25 °C for 30 min at 600 rpm, protected from direct light. Proteins were digested with 1:50 (m/m ratio) of trypsin and 1:100 of LysC for 16 h at 37 °C. Digestion was stopped by adding 0.5% (v/v) of trifluoroacetic acid. Further analysis was performed by EMBL Proteomics Core Facility in Heidelberg.

Digested peptides were concentrated and desalted using an OASIS HLB µElution Plate (Waters), according to manufacturer instructions. Crosslinked peptides were enriched using size exclusion chromatography[50]. In brief, desalted peptides were reconstituted with size exclusion chromatograph buffer (30% (v/v) acetonitrile (ACN) in 0.1% (v/v) trifluoroacetic acid (TFA)) and fractionated using a Superdex Peptide PC 3.2/30 column (GE) on a 1200 Infinity high-performance liquid chromatography system (Agilent) at a flow rate of 0.05 ml min$^{-1}$. Fractions eluting between 50–70 µl were evaporated to dryness and reconstituted in 30 µl 4% (v/v) ACN in 1% (v/v) FA.

Collected fractions were analyzed by liquid chromatography-coupled tandem mass spectrometry using an UltiMate 3000 RSLC nano liquid chromatography system (Dionex) fitted with a trapping cartridge (µ-Precolumn C18 PepMap 100, 5 µm, 300 µm × 5 mm, 100 Å) and an analytical column (nanoEase M/Z HSS T3 column 75 µm × 250 mm C18, 1.8 µm, 100 Å, Waters). Trapping was carried out with a constant flow of trapping solvent (0.05% trifluoroacetic acid in water) at 30 µl min$^{-1}$ onto the trapping column for 6 min. Subsequently, peptides were eluted and separated on the analytical column using a gradient composed of solvent A (3% dimethyl sulfoxide and 0.1% formic acid in water) and solvent B (3% dimethyl sulfoxide and 0.1% formic acid in acetonitrile) with a constant flow of 0.3 µl min$^{-1}$. The outlet of the analytical column was coupled directly to an Orbitrap Fusion Lumos (Thermo Scientific) mass spectrometer using the nanoFlex source.

The peptides were introduced into the Orbitrap Fusion Lumos via a Pico-Tip Emitter 360 µm × 20 µm; 10 µm tip (CoAnn Technologies) and an applied spray voltage of 2.1 kV, and the instrument was operated in positive mode. The capillary temperature was set at 275 °C. Only charge states of 4–8 were included. The dynamic exclusion was set to 30 s and the intensity threshold was 5 × 10$^4$. Full mass scans were acquired for a mass range 350–1,700 $m/z$ in profile mode in the orbitrap with resolution of 120,000. The AGC target was set to standard and the injection time mode was set to auto. The instrument was operated in data-dependent acquisition mode with a cycle time of 3 s between master scans and tandem mass spectrometry (MS/MS) scans were acquired in the Orbitrap with a resolution of 30,000, with a fill time of up to 100 ms and a limitation of 2 × 10$^5$ ions (AGC target). A normalized collision energy of 32 was applied. MS2 data were acquired in profile mode.

All data were analyzed using the cross-linking module in Mass Spec Studio v2.4.0.3524 (www.msstudio.ca, ref. 51). Parameters were set as follows: trypsin (K/R only), charge states 3–8, peptide length 7–50, percent $E$-value threshold of 50, mass spectrometry (MS) mass tolerance of 10 ppm, tandem mass spectrometry mass tolerance of 10 and elution width of 0.5 min. BS3 cross-links residue pairs were constrained to KSTY on one end and one of KSTY on the other. Identifications were manually validated, and cross-links with an $E$-value corresponding to <0.05% false discovery rate (FDR) were rejected. The data export from the Studio was filtered to retain only cross-links with a unique pair of peptide sequences and a unique set of potential residue sites.

Structural and functional analysis of the XL-MS data were performed with XiView[52].

## Reporting summary
Further information on research design is available in the Nature Portfolio Reporting Summary linked to this article.

## Data availability
Structural data have been deposited in PDB and EMDB under the following accession codes: PDB 8Q91 and EMD-18267 for the 20S U5 snRNP

core structure and PDB 8RC0 and EMD-19041 for the complete model of the 20S U5 snRNP. Other atomic coordinates used in this study for the comparisons purposes are available from the PDB under the following accession codes: 6QW6 for the U4/U6.U5 tri-snRNP, 6FF4 for the Bact complex; 6ID0 for the human ILS complex and 4I43 for AAR2–PRP8 complex. Quantitative proteomics and XL-MS data are provided as source data together with this manuscript. Other data and materials created within this study will be made available upon request. Source data are provided with this paper.

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

## Acknowledgements

We acknowledge M. Rettel, P. Haberkant and F. Stein from the European Molecular Biology Laboratory (EMBL) Proteomic Core Facility for their support with the mass spectrometry data acquisition and analysis; E. Marchal for the technical assistance with the KO cell line preparation; M. Pelosse and A. Aubert from the EMBL Grenoble EEF platform for the assistance with cell culture. We acknowledge support of the EM Facilities at EMBL Grenoble and Heidelberg. ESRF for the provision of CM01 beamtime and Michael Hons for assistance with the preliminary data collection; Mylene Pezet from the IAB Grenoble Flow Cytometry Platform for the assistance with cell sorting; A. Peuch and the EMBL Grenoble IT team for the support with high-performance computing. This project has received funding from the EMBL, the European Research Council under the European Union's Horizon 2020 research and innovation program (grant agreement no. 950278, awarded to W.P.G.) and Human Frontiers Science Program (HSFP Postdoctoral Fellowship LT000383/2018-L awarded to D.P.).

## Author contributions

D.P. and J.P.-M. created CD2BP2 cell lines and established U5 snRNP purification conditions. D.P. and W.P.G. performed initial cryo-EM characterization. I.B. purified 20S U5 snRNP, performed TMG pull-down and XL-MS experiments. S.S. and I.B. prepared cryo-EM grids and analyzed XL-MS data. S.S. collected and processed cryo-EM data. S.S., J.Z. and W.P.G. analyzed the structure and built and refined atomic models. A.F., J.T. and S.L. performed additional biochemical assays. W.P.G. initiated and supervised the project and wrote the manuscript with input from all authors.

## Funding

## Competing interests

The authors declare no competing interests.

## Additional information

**Extended data** is available for this paper at https://doi.org/10.1038/s41594-024-01250-5.

**Correspondence and requests for materials** should be addressed to Wojciech P. Galej.

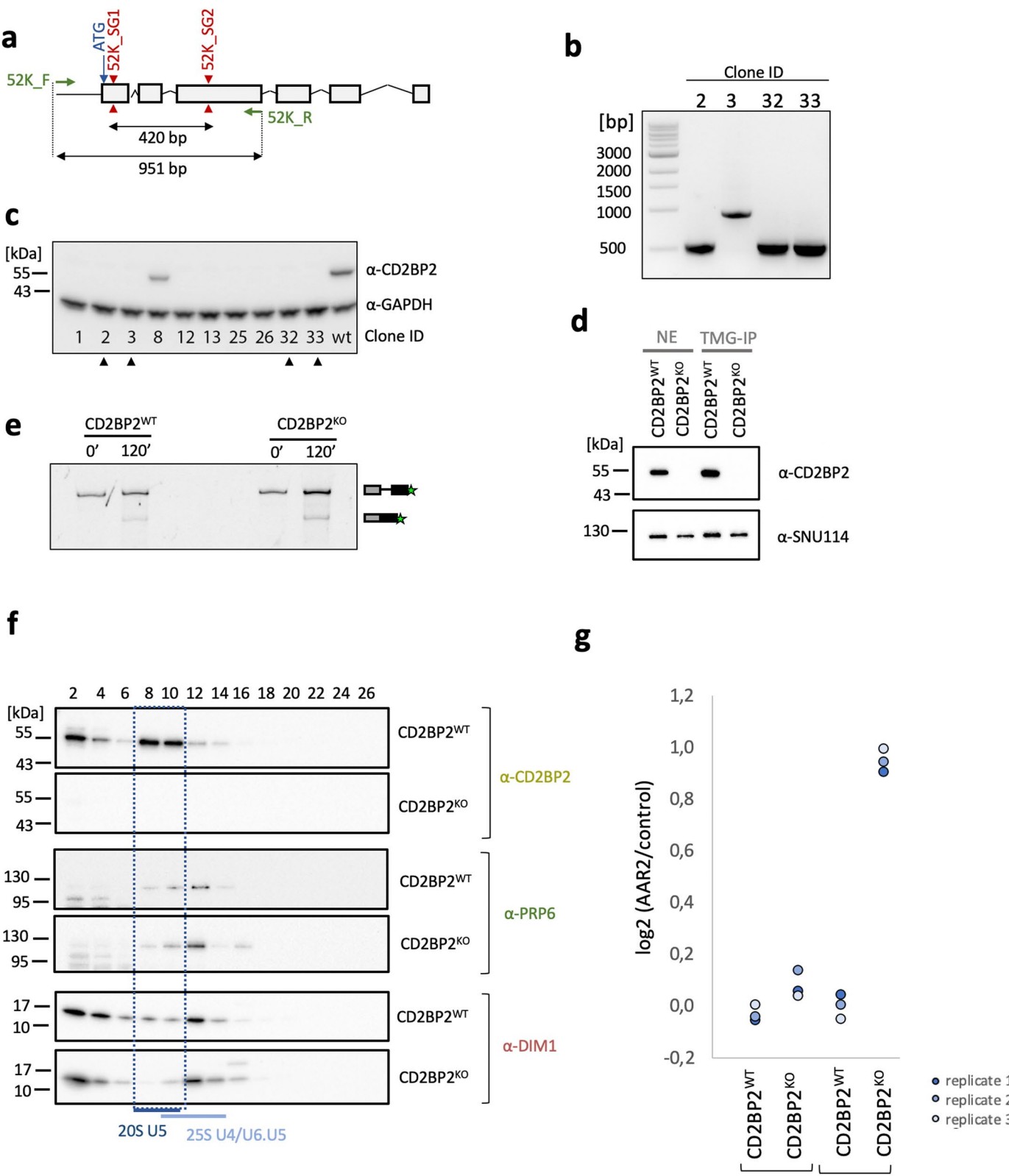

**Extended Data Fig. 1 | CD2BP2 knock out and its impact on pre-mRNA splicing and snRNPs assembly. a**, experimental design used in CRISPR/Cas9-mediated CD2BP2[KO] generation; **b**, genotyping of the CD2BP2[KO] clones using PCR as indicated in panel a; **c**, Western blot analysis of the knockout cell lines with anti-CD2BP2 antibodies and GAPDH used as a loading control; **d**, Western blot of the NE and TMG pull-down fractions used in Fig. 1 a probed for the presence of CD2BP2 and SNU114; **e**, *In vitro* splicing assay of the AdML-M3 pre-mRNA substrate in the nuclear extract prepared from WT or CD2BP2[KO] cell lines; **f**, Western blot of the glycerol gradient fractions probed with U5 snRNP-specific antibodies, showing depletion of DIM1 from 20 S U5 snRNP in CD2BP2 KO condition; **g**, Quantitative mass spectrometry measurement of the AAR2 abundance in the input nuclear extracts (NE) and TMG pull-down fractions (TMG-PD) from WT and CD2BP2 knock-out cell lines. Experiments in b-e were performed in single biological replicates, while d-g were done in triplicates with consistent outcomes.

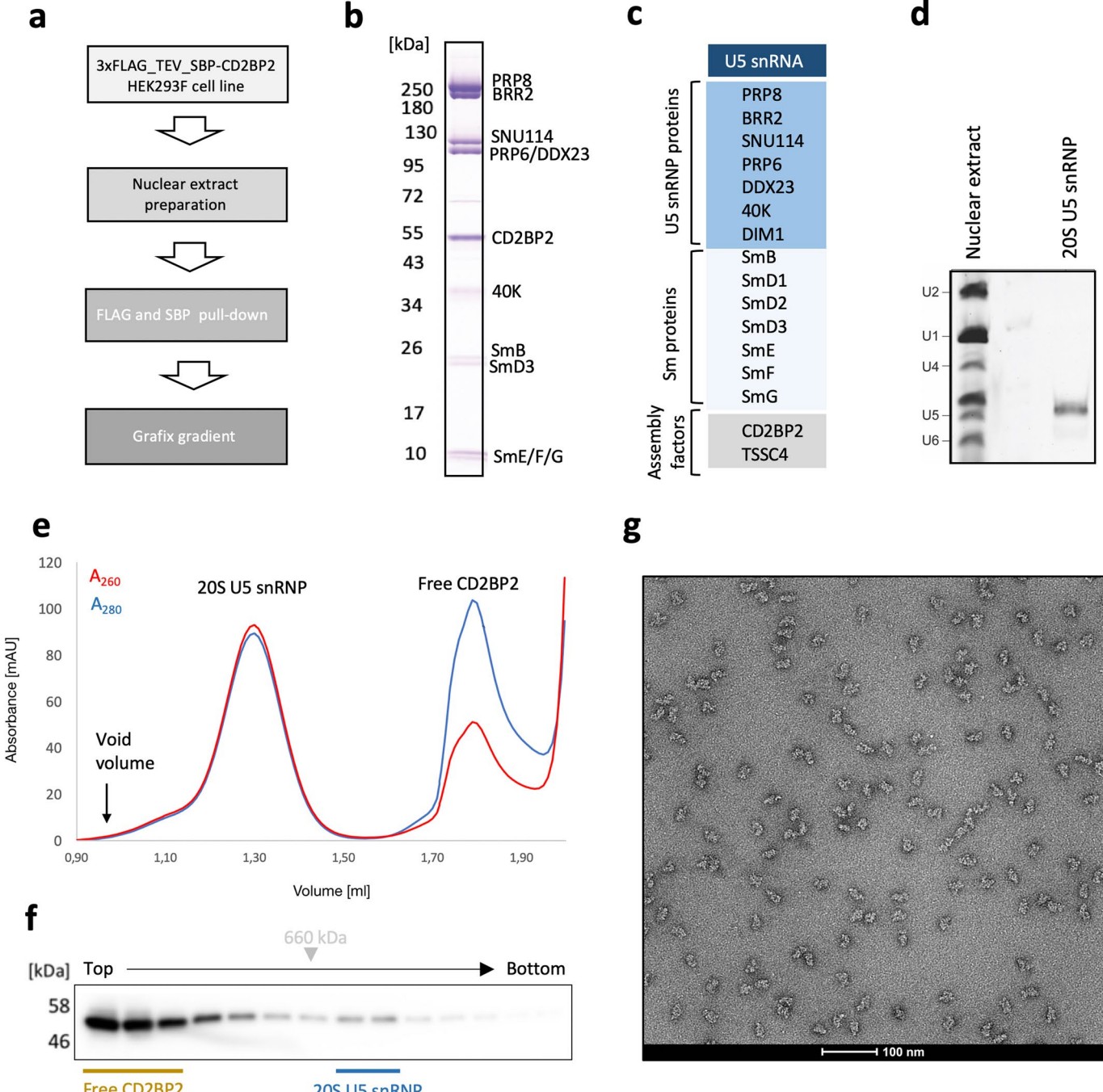

**Extended Data Fig. 2 | Purification of the 20S U5 snRNP from HEK293F cells.** **a**, experimental workflow used in sample preparation; **b**, SDS-PAGE analysis of the SBP eluates used subsequently for the Grafix gradient; **c**, a list of proteins present in the 20S U5 snRNP preparation; **d**, UREA-PAGE analysis of the RNAs extracted from SBP eluate of the CD2BP2-purified sample or NE used as a control. RNAs were stained with SYBR Gold and detected using Bio-Rad Chemidoc; **e**, 20S U5 snRNP sample analysed on the Superose 6 3.2/300 size exclusion chromatography column; **f**, the same sample analysed on a non-cross-linking 10–30% glycerol gradient and detected by Western blot with anti-FLAG antibodies. The migration of the thyroglobulin size marker is indicated with the grey arrow; **g**, a typical micrograph of negatively stained Grafix fractions used subsequently for cryo-EM analysis. Experiments in panels b,d,g and f, were performed in triplicates with consistent outcomes.

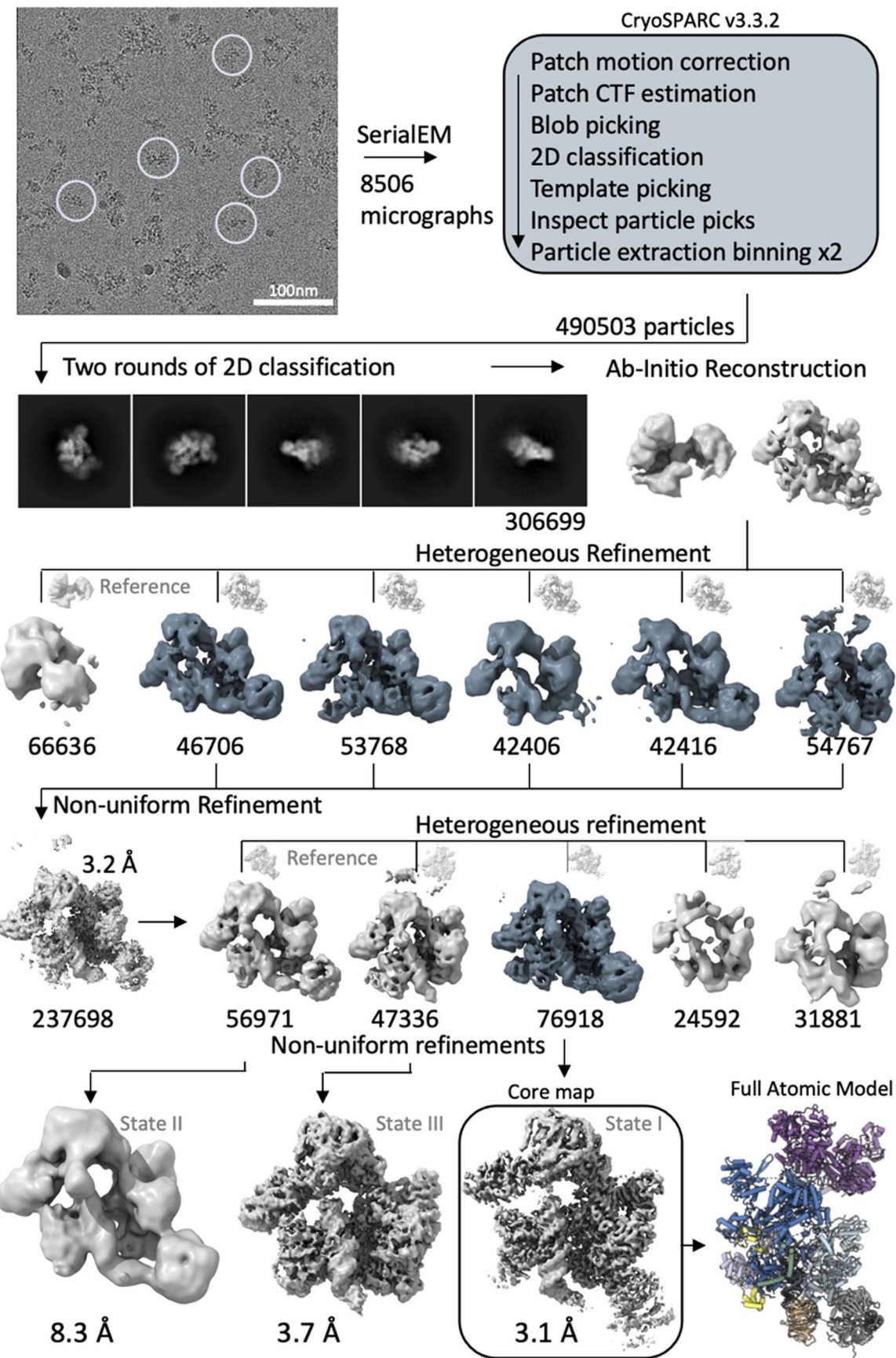

**Extended Data Fig. 3 | Cryo-EM data processing chart.** Processing chart for the 20S U5 snRNP. The dataset contained compositionally heterogeneous snRNP, which were separated by heterogeneous refinement. This was then followed by non-uniform refinement. References are depicted in grey. Numbers indicate particle numbers unless otherwise stated. Extended Data Fig. 6 contains a more detailed comparison of the final three reconstructions.

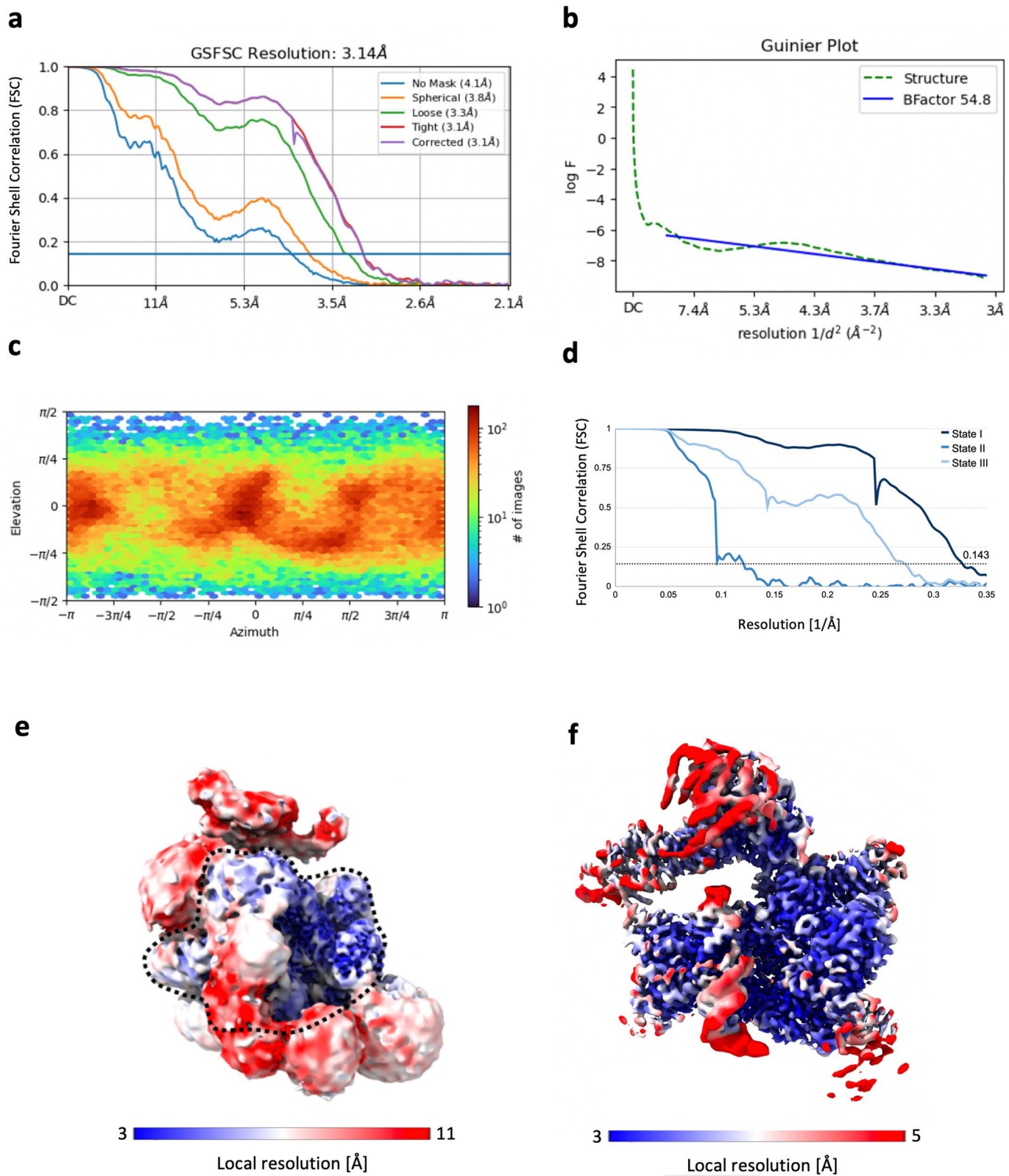

**Extended Data Fig. 4 | Global and local resolution analysis of the cryo-EM reconstructions obtained in this study.** Graphs based on the 3.1 Å, determined with cryoSPARC v3.3.2 unless otherwise indicated. **a**, Fourier Shell Correlation (FSC) of the final 3.1 Å cryo-EM map with different masks; **b**, Guinier plot used to determine the b-factor of 54.8 Å²; **c**, Angular distribution shows no major preferential orientation; **d**, FSC curves of the final three reconstructions; **e**, Local resolution plotted on the isosurface of the locally filtered state I map at a low contour level. The core of the particle shown in panel **f**, is indicated with a dotted line.

**a**

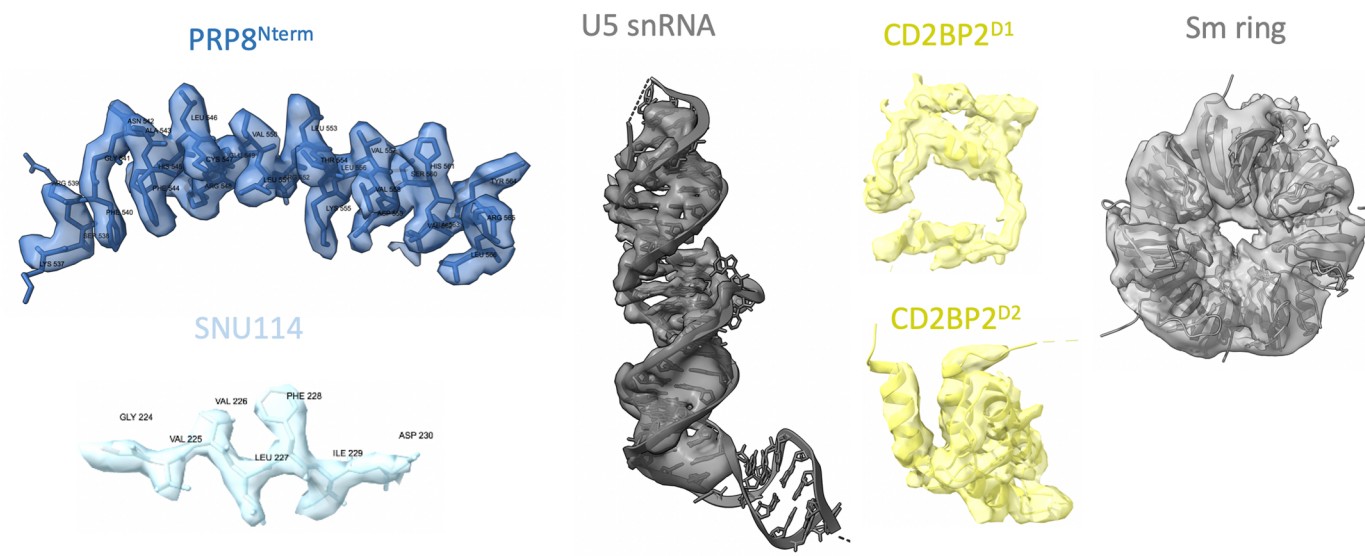

**b**

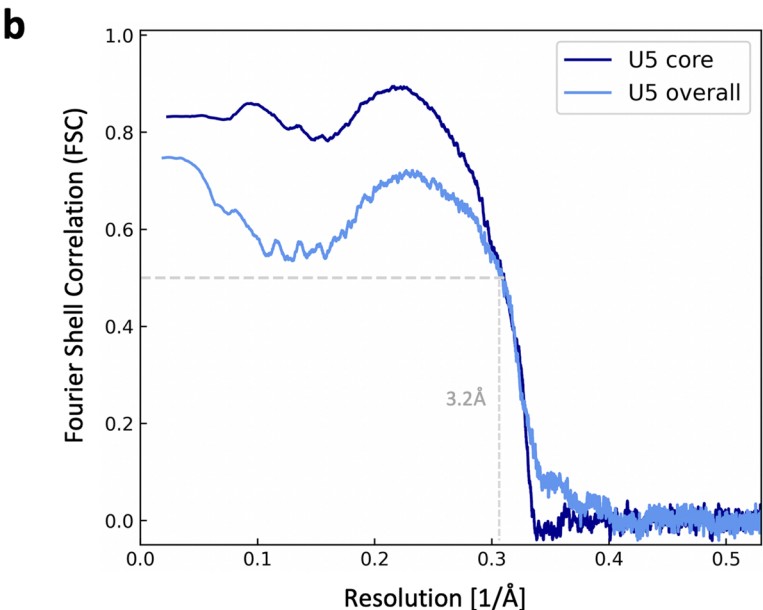

**Extended Data Fig. 5 | Examples of the atomic model fitting into the cryo-EM density of the 20S U5 snRNP. a**, Representative examples of PRP8 and SNU114, U5 RNA and CD2BP2 areas resolved at high-resolution. The Sm ring is an example of a low-resolution fitting. **b**, Fourier Shell Correlation (FSC) of the model vs map shows good agreement up to 3.2 Å (FSC = 0.5 cut-off).

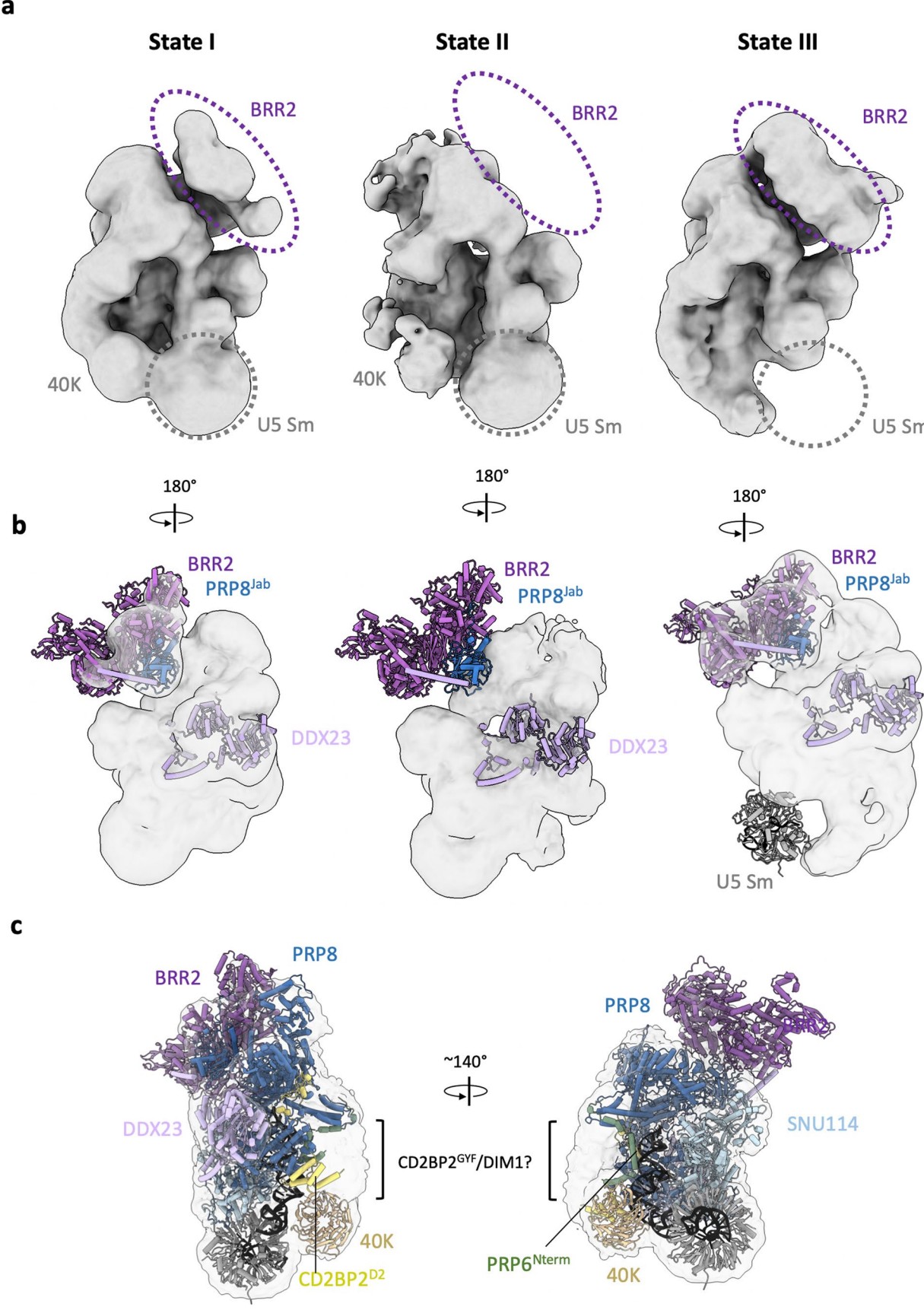

**Extended Data Fig. 6 | See next page for caption.**

**Extended Data Fig. 6 | Three major compositional states are present in the reconstruction of the 20S U5 snRNP. a**, Low-pass filtered reconstructions of the three states present in the reconstruction highlighting the key differences between them; **b**, Atomic models of the highlighted components fitted into the corresponding maps. State I represents the fully assembled complex referred to as the 20S U5 snRNP; state II has overall very similar architecture, but no clear density for BRR2 and DDX23 was observed; state III misses Sm ring and 40K and most likely represents broken particles. A comparison of state I and state II shows that BRR2 and DDX23 stabilise each other, as their presence is largely

correlated. It has been previously shown that DDX23 binding to BRR2 is mutually exclusive with TSSC4 (ref. [9]). Therefore, the absence of DDX23/BRR2 in one of the classes could, in principle, represent a state where TSSC4 is bound to BRR2 and prevents its DDX23-mediated stable docking to the body of the complex. However, we could not see density for TSSC4 in any of these reconstructions even though putative TSSC4-PRP8 interface could be predicted with AlphaFold2; **c**, Unassigned density located near 40K, CD2BP2 and PRP8 most likely represents DIM/CD2BP2^GYF domain.

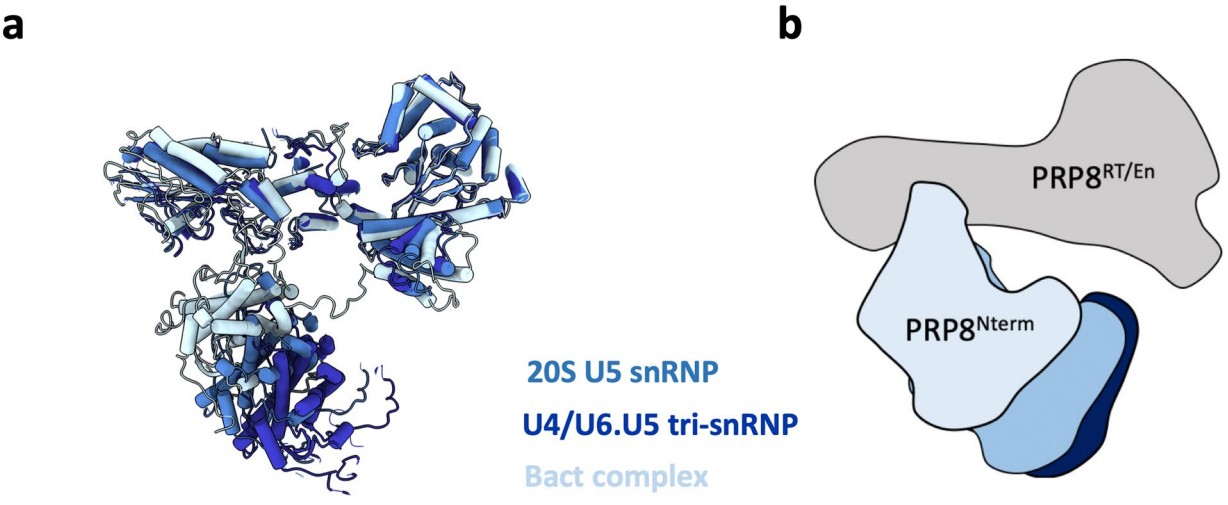

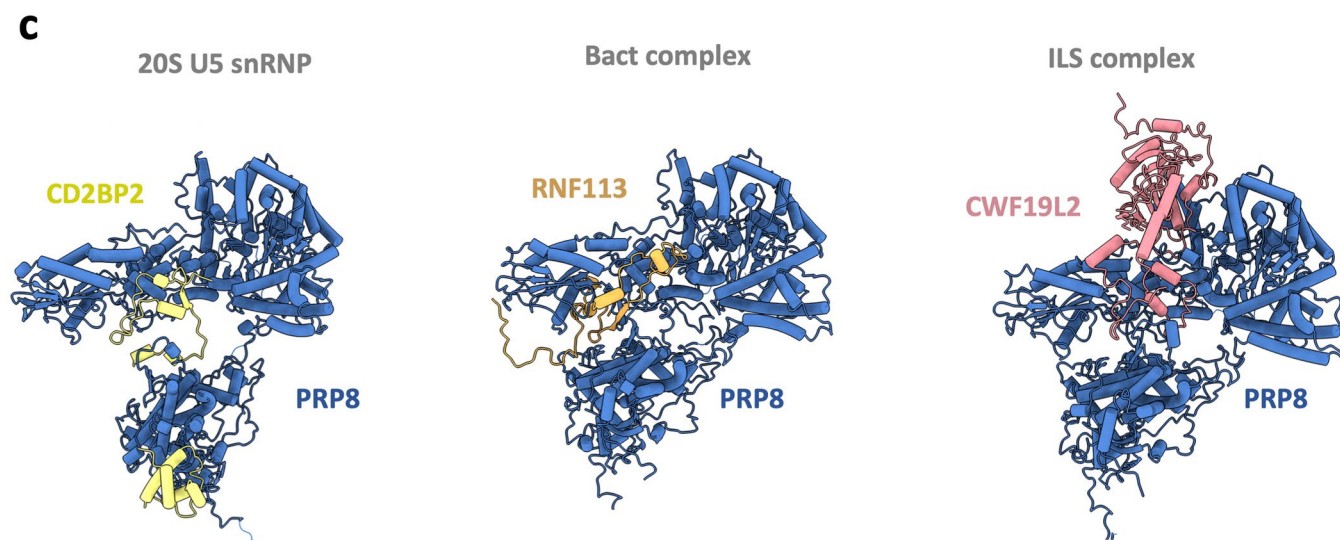

**Extended Data Fig. 7 | PRP8 conformation in the 20S U5 snRNP.**
**a**, superposition of PRP8 from 20S U5 snRNP structure (this work), U4/U6.U5 tri-snRNP[21] (PDB:6QW6) and Bact spliceosome[24] (PDB:6FF4) showing the movement of the PRP8[Nterm] with respect to PRP8[RT/EN] in different splicing complexes. The position of this domain in 20S U5 snRNP is likely stabilised by CD2BP2[D1];

**b**, a cartoon representation highlighting the movement described in panel a.
**c**, CD2BP2 binding surface on PRP8[RT/EN] domain is occupied by different factors in other splicing complexes: RNF113 in Bact spliceosome[24] and CW19L2 in post-splicing ILS complex[25].

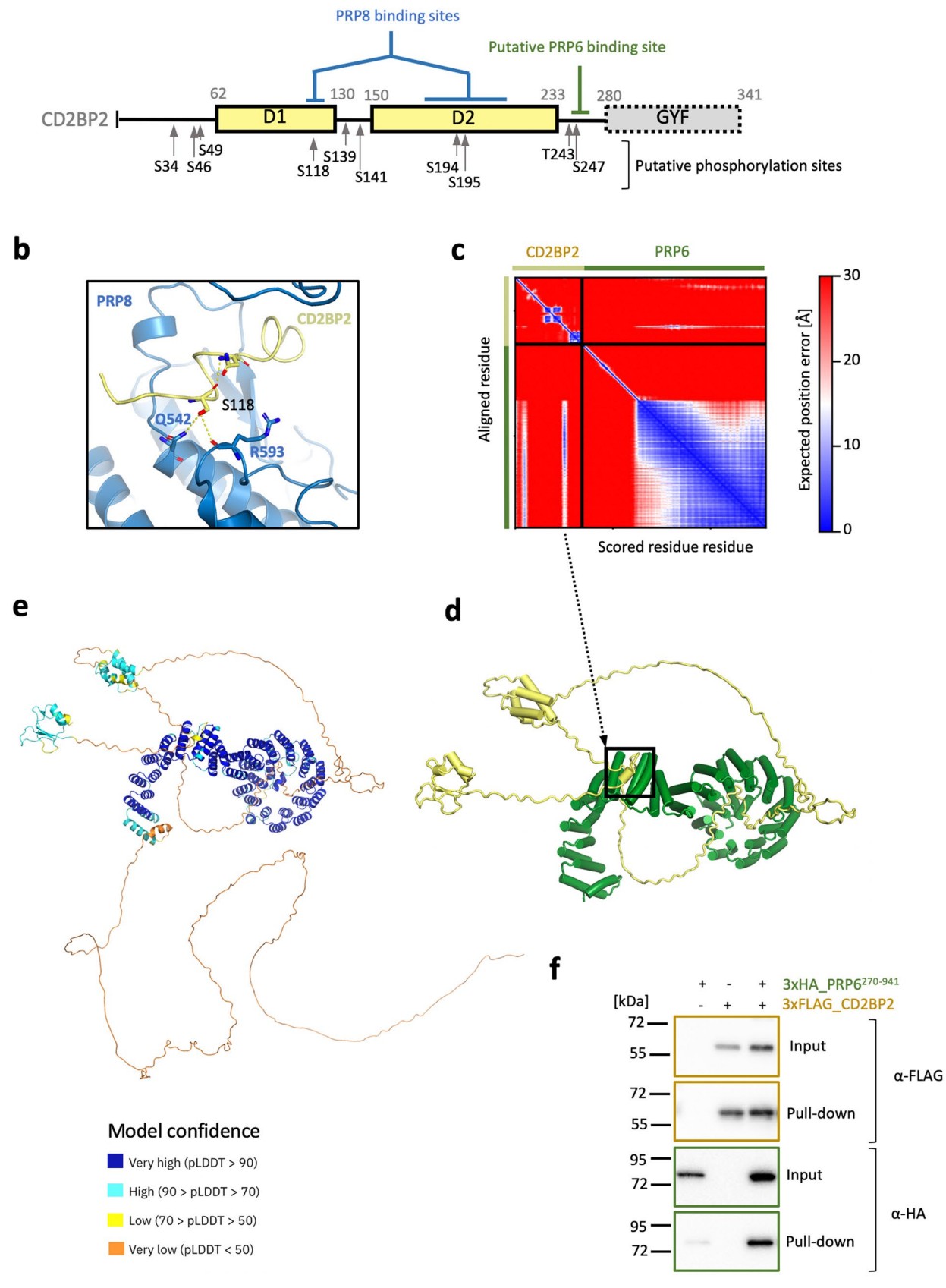

**Extended Data Fig. 8 | See next page for caption.**

**Extended Data Fig. 8 | Putative CD2BP2-PRP6 interface and phosphorylation sites in CD2BP. a**, putative phosphorylation sites in CD2BP2 detected in high-throughput experiments[53,54] mapped on the primary sequence representation; **b**, Interaction network of CD2BP2 Serine 118, a putative phosphorylation site that is well resolved in the structure; **c** and **d**, a cartoon representation and the PAE plot of the AlphaFold2 model of the CD2BP2-PRP6 binary complex; **e**, AlphaFold2 model of the full-length PRP6-CD2BP2 complex coloured based on pLDDT score (prediction confidence metric); **f**, Western blot analysis of the FLAG-tag pull-down experiment from HEK293T cells co-expressing 3xFLAG-CD2BP2 and 3xHA-PRP6[TPR], confirming direct interaction between the two proteins. Experiment in panel f was performed in a single biological replicate.

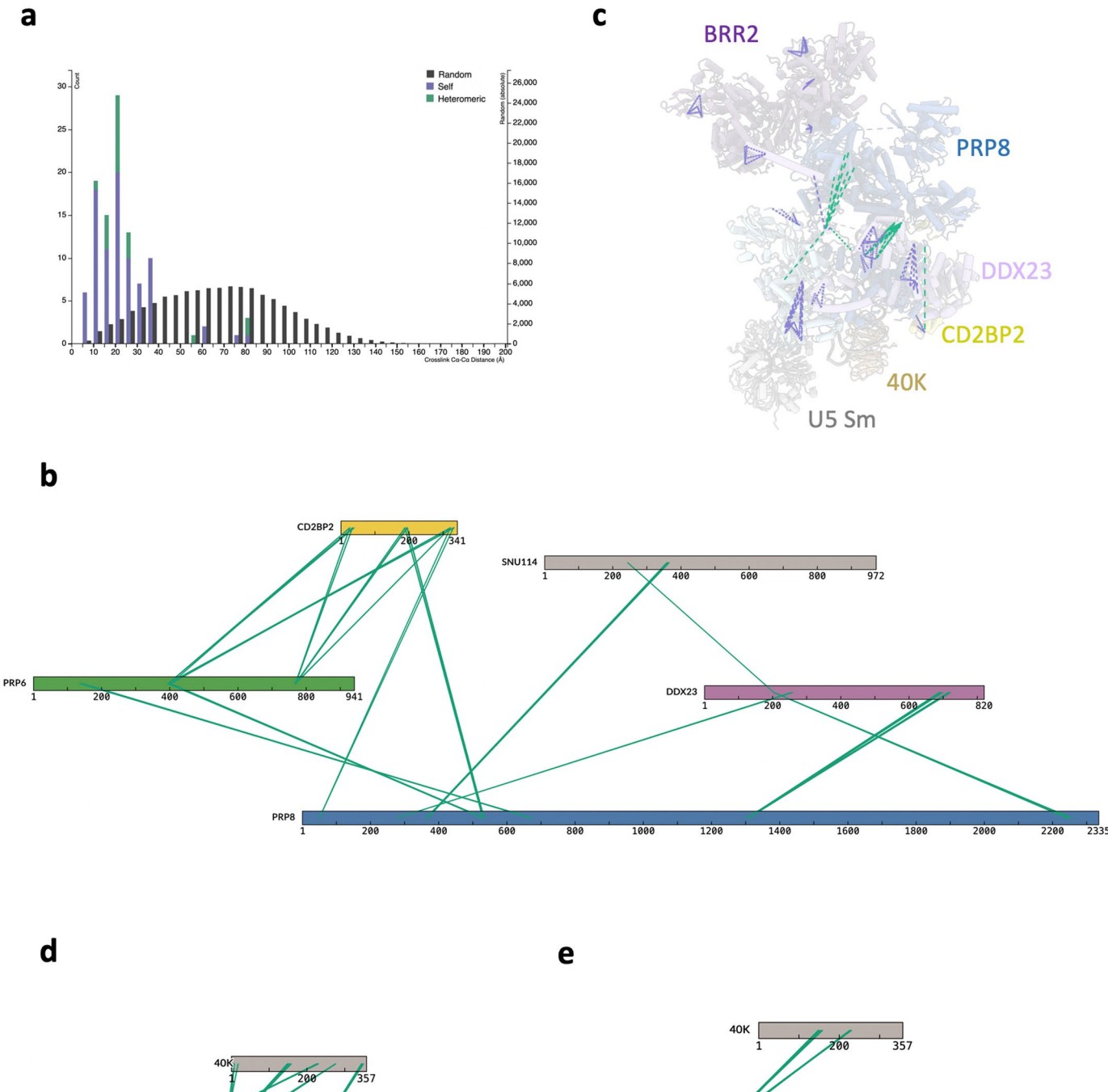

**Extended Data Fig. 9 | BS3 cross-linking and mass spectrometry analysis of the 20S U5 snRNP. a**, distribution of the Cα-Cα distances mapped on the core U5 snRNP structure. The majority of the inter- and intra-molecular cross-links fall within 35 Å distance, consistent with the chemical nature of the cross-linker; **b**, visualisation of the high-confidence cross-links (Match score > 180) with respect to their position in their target proteins; **c**, mapping of the high-confidence cross-links onto the structure of the 20S U5 snRNP; **d** and **e**, unfiltered cross-links from two independent experiments (0.25 mM and 1 mM BS3) mapped on the sequences of CD2BP2, DIM1, PRP6 and 40K. These contacts support the model, wherein C-terminus of CD2BP2 and DIM1 are located near 40K. PRP6 TPR repeats appear to form multiple cross-links with CD2BP2, consistent with the AF2 model and biochemical data (Extended Data Fig. 8). XL-MS data was analysed and visualized in the xiView[52].

**Extended Data Table 1 | Solution mass spectrometry analysis of the purified 20 S U5 snRNP**

| Protein | Top3[1] | Unique peptides | Total spectral count |
|---|---|---|---|
| PRP8 | 10,6710814 | 176 | 2781 |
| SNU114 | 10,5225461 | 72 | 1526 |
| BRR2 | 10,419635 | 183 | 3299 |
| 40K | 10,4013987 | 27 | 406 |
| PRP6 | 10,3256526 | 67 | 684 |
| DDX23 | 10,2754863 | 57 | 341 |
| SmF | 10,1563242 | 9 | 64 |
| SmD3 | 9,9275086 | 4 | 24 |
| SmD1 | 9,9013988 | 6 | 91 |
| CD2BP2 | 9,84841526 | 7 | 1270 |
| DIM1 | 9,84199222 | 12 | 101 |
| SNRPD2 | 9,79106248 | 8 | 18 |
| SmE | 9,45082645 | 5 | 26 |
| TSSC4 | 9,38630908 | 12 | 31 |
| HSPA8[2] | 8,54658185 | 14 | 31 |
| SH2D4A[2] | 8,13403687 | 6 | 7 |
| TUBB[2] | 8,10539485 | 2 | 15 |
| KRT9[2] | 7,45557642 | 2 | 2 |

Proteins in the table were sorted based on their Top3 values in a descending order. [1]The Top3 value is the average log10 MS1 intensity of the three most abundant peptides for each protein and serves as an estimator for the average abundance. [2]These proteins are typical contaminants present in the preparation.

# Reporting Summary

## Statistics

For all statistical analyses, confirm that the following items are present in the figure legend, table legend, main text, or Methods section.

| n/a | Confirmed | |
|---|---|---|
| ☒ | ☐ | The exact sample size (*n*) for each experimental group/condition, given as a discrete number and unit of measurement |
| ☒ | ☐ | A statement on whether measurements were taken from distinct samples or whether the same sample was measured repeatedly |
| ☐ | ☒ | The statistical test(s) used AND whether they are one- or two-sided *Only common tests should be described solely by name; describe more complex techniques in the Methods section.* |
| ☒ | ☐ | A description of all covariates tested |
| ☒ | ☐ | A description of any assumptions or corrections, such as tests of normality and adjustment for multiple comparisons |
| ☒ | ☐ | A full description of the statistical parameters including central tendency (e.g. means) or other basic estimates (e.g. regression coefficient) AND variation (e.g. standard deviation) or associated estimates of uncertainty (e.g. confidence intervals) |
| ☐ | ☒ | For null hypothesis testing, the test statistic (e.g. *F*, *t*, *r*) with confidence intervals, effect sizes, degrees of freedom and *P* value noted *Give P values as exact values whenever suitable.* |
| ☒ | ☐ | For Bayesian analysis, information on the choice of priors and Markov chain Monte Carlo settings |
| ☒ | ☐ | For hierarchical and complex designs, identification of the appropriate level for tests and full reporting of outcomes |
| ☒ | ☐ | Estimates of effect sizes (e.g. Cohen's *d*, Pearson's *r*), indicating how they were calculated |

*Our web collection on statistics for biologists contains articles on many of the points above.*

## Software and code

Policy information about availability of computer code

| Data collection | SerialEM, Mass Spec Studio v2.4.0.3524 |
|---|---|
| Data analysis | CryoSPARC v3.3; UCSF Chimera 1.5; Coot v0.9.8.5; Refmac Servalcat v5.8.0267; Pymol 2.5.5; PHENIX 1.20.1_4487; XiView; cryoSPARC v3.3 |

For manuscripts utilizing custom algorithms or software that are central to the research but not yet described in published literature, software must be made available to editors and reviewers. We strongly encourage code deposition in a community repository (e.g. GitHub). See the Nature Portfolio guidelines for submitting code & software for further information.

## Data

Policy information about availability of data

All manuscripts must include a data availability statement. This statement should provide the following information, where applicable:
- Accession codes, unique identifiers, or web links for publicly available datasets
- A description of any restrictions on data availability
- For clinical datasets or third party data, please ensure that the statement adheres to our policy

Structural data has been deposited in PDB and EMDB under the following accession codes: PDB 8Q91 and EMD-18267 for the 20S U5 snRNP core structure and PDB 8RC0 and EMD-19041 for the complete model of the 20S U5 snRNP. Other atomic coordinates used in this study for the comparisons purposes are available from the PDB under the following accession codes: 6QW6 for the U4/U6.U5 tri-snRNP, 6FF4 for the Bact complex; 6ID0 for the human ILS complex and 4I43 for Aar2-Prp8 complex. Other data and materials created within this study will be made available upon request.

## Research involving human participants, their data, or biological material

Policy information about studies with human participants or human data. See also policy information about sex, gender (identity/presentation), and sexual orientation and race, ethnicity and racism.

| | |
|---|---|
| Reporting on sex and gender | N/A |
| Reporting on race, ethnicity, or other socially relevant groupings | N/A |
| Population characteristics | N/A |
| Recruitment | N/A |
| Ethics oversight | N/A |

Note that full information on the approval of the study protocol must also be provided in the manuscript.

# Field-specific reporting

Please select the one below that is the best fit for your research. If you are not sure, read the appropriate sections before making your selection.

☒ Life sciences ☐ Behavioural & social sciences ☐ Ecological, evolutionary & environmental sciences

For a reference copy of the document with all sections, see nature.com/documents/nr-reporting-summary-flat.pdf

# Life sciences study design

All studies must disclose on these points even when the disclosure is negative.

| | |
|---|---|
| Sample size | cryo-EM data sample size was determined by the data processing outcome yielding a desired resolution. |
| Data exclusions | Some of the cryo-EM data was excluded during classification procedures following common practices in this field. |
| Replication | Quantitative proteomics experiments were performed in three biological replicates. |
| Randomization | Two half-datasets used to calculate gold-standard FSC curves were assigned randomly. |
| Blinding | Investigators were blinded with respect to half-datasets assignments. |

# Reporting for specific materials, systems and methods

We require information from authors about some types of materials, experimental systems and methods used in many studies. Here, indicate whether each material, system or method listed is relevant to your study. If you are not sure if a list item applies to your research, read the appropriate section before selecting a response.

### Materials & experimental systems

| n/a | Involved in the study |
|---|---|
| ☐ | ☒ Antibodies |
| ☐ | ☒ Eukaryotic cell lines |
| ☒ | ☐ Palaeontology and archaeology |
| ☒ | ☐ Animals and other organisms |
| ☒ | ☐ Clinical data |
| ☒ | ☐ Dual use research of concern |
| ☒ | ☐ Plants |

### Methods

| n/a | Involved in the study |
|---|---|
| ☒ | ☐ ChIP-seq |
| ☒ | ☐ Flow cytometry |
| ☒ | ☐ MRI-based neuroimaging |

## Antibodies

| | |
|---|---|
| Antibodies used | Anti-2,2,7-Trimethylguanosine Mouse antibodies, K121, Agarose Conjugate, Merck NA02A<br>CD2BP2 (Sigma, HPA061309), 1:5000<br>DIM1 (Proteintech, 27646-1-AP), 1: 3000<br>PRP6 (Invitrogen, PA5-61428), 1: 2000 |

SNU114/EFTUD2 (Invitrogen, PA5-96559) 1:5000
goat anti-rabbit IgG H&L, HRP (Abcam, ab205718), 1:5000
Anti-FLAG M2-Peroxidase (Sigma, A8592), 1: 5000
 anti-GAPDH (Invitrogen MA515738HRP), 1:5000
 anti-HA-Tag F-7 (SantaCruz, sc-7392), 1:5000

Validation
validation performed by manufacturers

## Eukaryotic cell lines

Policy information about cell lines and Sex and Gender in Research

Cell line source(s)
HEK293T (ATCC),Freestyle 293 cells (ThermoFisher)

Authentication
cell lines were not authenticated

Mycoplasma contamination
not tested

Commonly misidentified lines
(See ICLAC register)
*Name any commonly misidentified cell lines used in the study and provide a rationale for their use.*

## Plants

Seed stocks
*Report on the source of all seed stocks or other plant material used. If applicable, state the seed stock centre and catalogue number. If plant specimens were collected from the field, describe the collection location, date and sampling procedures.*

Novel plant genotypes
*Describe the methods by which all novel plant genotypes were produced. This includes those generated by transgenic approaches, gene editing, chemical/radiation-based mutagenesis and hybridization. For transgenic lines, describe the transformation method, the number of independent lines analyzed and the generation upon which experiments were performed. For gene-edited lines, describe the editor used, the endogenous sequence targeted for editing, the targeting guide RNA sequence (if applicable) and how the editor was applied.*

Authentication
*Describe any authentication procedures for each seed stock used or novel genotype generated. Describe any experiments used to assess the effect of a mutation and, where applicable, how potential secondary effects (e.g. second site T-DNA insertions, mosiacism, off-target gene editing) were examined.*

