## [Peer Review File · Nature Structural & Molecular Biology]

Peer Review Information

Manuscript Title: Structure of the human 20S U5 snRNP

Corresponding author name(s): Wojciech Galej

Reviewer Comments & Decisions:

Decision Letter, initial version:

Message: 5th Oct 2023

Dear Dr. Galej,

Thank you again for submitting your manuscript "Structure of the human 20S U5 snRNP". I apologise for the delay in responding, which resulted from the difficulty in timely obtaining suitable referee reports. Nevertheless, we now have comments (below) from the 2 reviewers who evaluated your paper. In light of these reports, we remain interested in your study and would like to see your response to the comments of the referees, in the form of a revised manuscript.

You will see that the experts find the novel structural data, and the potential mechanistic implications imparted by them, interesting and deem that they will be helpful to the splicing field and will spark further studies. However, both referees voice technical and mechanistic concerns that need to be addressed in revised manuscript. More specifically, both experts point to inconsistencies and/or insufficiently clear resolution in the reported structural data that must be fixed. Moreover, Reviewer #3 poses a few mechanistic questions that, if answered, would significantly elevate the mechanistic value of the manuscript. Finally, both experts request additional clarifications at places, further discussion and contextualisation of the findings, with guidance that we ask you to please follow.

Please be sure to address/respond to all concerns of the referees in full in a point-by-point response and highlight all changes in the revised manuscript text file. If you have comments that are intended for editors only, please include those in a separate cover letter.

We expect to see your revised manuscript within 6 weeks. If you cannot send it within this time, please contact us to discuss an extension; we would still consider your revision, provided that no similar work has been accepted for publication at NSMB or published elsewhere.

Reporting Summary:

Data availability: this journal strongly supports public availability of data. All data used in accepted papers should be available via a public data repository, or alternatively, as

Supplementary Information. If data can only be shared on request, please explain why in your Data Availability Statement, and also in the correspondence with your editor. Please note that for some data types, deposition in a public repository is mandatory - more information on our data deposition policies and available repositories can be found below: <https://www.nature.com/nature-research/editorial-policies/reporting-standards#availability-of-data>

Nature Structural & Molecular Biology is committed to improving transparency in authorship. As part of our efforts in this direction, we are now requesting that all authors identified as 'corresponding author' on published papers create and link their Open Researcher and Contributor Identifier (ORCID) with their account on the Manuscript Tracking System (MTS), prior to acceptance. This applies to primary research papers only. ORCID helps the scientific community achieve unambiguous attribution of all scholarly contributions. You can create and link your ORCID from the home page of the MTS by clicking on 'Modify my Springer Nature account'. For more information please visit [visit www.springernature.com/orcid](http://www.springernature.com/orcid).

[redacted]

Sincerely,

Dimitris Typas
Associate Editor
Nature Structural & Molecular Biology
ORCID: 0000-0002-8737-1319

Referee expertise:

Referee #2: structural biology of RNA-built macromolecular complexes

Referee #3: structural biology of pre-mRNA splicing

Reviewers' Comments:

Reviewer #2:

Remarks to the Author:

The 20S U5 small nuclear ribonucleoprotein particle (snRNP) is a complex consisting of 17 subunits, comprised of both RNA and protein components. It serves as a precursor to the U4/U6.U5 tri-snRNP, which is a crucial component of the pre-catalytic spliceosome. Several assembly/recycling factors associating with the tri-snRNP have been identified (AAR2, CD2BP2, TSSC4, ZNHIT2 as well as U4/U6 annealing factor SART3) but their precise role remains mechanistically elusive. In this brief communication, Schneider, Brandina, Peter et al. describe the role of CD2BP2 in 20S U5 snRNP biogenesis using biochemical and cryo-electron microscopy analyses.

Using quantitative mass spectrometry, the authors compared protein composition in WT and homozygous CD2BP2KO cell lines and showed that levels of U5 and U4/U6.U5 tri-snRNP are affected in the absence of CD2BP2, highlighting a role of CD2BP2 in 20S U5 snRNP assembly or recycling. Next, the authors analyzed 20S U5 snRNPs using single particle cryo-EM. Three structures of 20S U5 snRNP were determined, and only one contains most expected components (State I). CD2BP2 was observed in the structure interacting with PRP8RT/EN and PRP8Nterm, stabilizing their relative conformation which differs from the one observed in the tri-SNP. CD2BP2 also occupies mutually exclusive binding sites on the PRP8 surface that other splicing factors engage during splicing. Finally, CD2BP2 is known to bind DIM1 and PRP6 (core) but these factors are not visible in the structure. Based on their structural data, the authors propose a dual mechanism for CD2BP2: First, tethering of PRP6 and DIM1 by CD2BP2 facilitates their recruitment to the 20S U5 snRNP, both splicing factors being critical for the tri-snRNP formation. Second, CD2BP2 acts as a placeholder preventing premature association of splicing factors to PRP8.

The paper is concise and well written. The figures are clear. Overall, this work provides new insights into the role of CD2BP2 in 20S U5 snRNP biogenesis, which will be of general interest for the splicing field and the RNA community. Publication of this work in NSMB is supported, provided that the following points are addressed:

Major points

1. After inspection of the provided atomic coordinates and cryo-EM maps, a couple of points should be addressed to improve the quality of the published structural data:
 - Any ligand without defined cryo-EM density to support their presence should be removed (e.g. IHP).
 - Side chains in low quality or low-resolution regions should be fully trimmed to the C-beta position (BRR2, DDX23, Sm proteins, etc).
 - The atomic model for the "complete" 20S U5 snRNP seems to be a composite of the

refined "core" atomic model and additional rigid body fitted proteins (BRR2, DDX23, etc). This is okay but B-factors (ADP) must be refined in the local-filtered map so that they reflect the local resolution from the EM data.

- In the "core" atomic model, chain D Arg 73 is floating in space.

2. There are inconsistencies between the provided coordinates, refinement statistics and FSC curves presented in the Extended Data figures:

- Extended Data Fig. 4: panel a shows the FSC curve of "3.1Å" (which is "State I" based on Extended Data Fig. 3 and should be indicated as such in the legend) while panel d presents FSC curves for 3 determined structures. FSC curves for State I in both panels a and d don't match.

- Extended Data Figure 5: panel b, model-to-map FSC should be plotted to at least Nyquist frequency.

- Table 2: the refinement statistics and reported refinement software (Refmac) do not match the header of the provided PDB files (Phenix). The indicated Clashscore is also relatively high for this kind of resolution and does not match the PDB header.

Overall, the authors should recheck the provided processing and refinements data for consistency.

3. The authors should provide a description of the Western blotting and In-vitro splicing assay protocols in the Methods section.

Minor points

1. The authors mention TSSC4 being part of the immature/recycled 20S U5 snRNP. Additionally, this factor is present in the purified 20S U5 snRNP (mass spectrometry analysis). However, TSSC4 was not identified nor visualized in the cryo-EM maps. This is discussed in the legend of Extended Fig. 6 but this is actually an important point and could be worth including in the main text if space permits.

2. Extended Data Fig. 8: panel d, the two protein chains should be colored based on the AlphaFold pLDDT score so that the reader can appreciate the confidence of the predicted protein interfaces.

6. Typo: top of page 2, "Extended Data Fig. 87".

Reviewer #3:

Remarks to the Author:

The U4/U6.U5 tri-snRNP is the largest building block of the spliceosome. It assembles from a 20S U5 snRNP and a U4/U6 di-snRNP particle. A hallmark of the 20 S U5 snRNP is the protein CD2BP2 that is believed to play a role in the biogenesis of the tri-snRNP. While a cryo-EM structure of the human tri-snRNP has been available for several years, a high-resolution structure of the 20S U5 snRNP has been missing. In this manuscript, Galej and coworkers succeeded in reconstructing a 3.1 Å cryo-EM structure of a 20 S U5 snRNP particle, which was purified using a tagged CD2BP2 protein. The key observation in this manuscript is that CD2BP2's N-terminal D1 domain straddles the PRP8 Large and N domains and thus stabilizes the orientation of these two PRP8 domains towards each other. Moreover, the binding of CD2BP2-D1 to PRP8 is mutually exclusive with the binding of the U5 protein DIM1, nicely explaining why the authors could not map DIM 1 in their structure. As DIM1 is well represented in the MS analysis of the purified U5 snRNP, the

authors assume that DIM1 may be tethered to the C-terminal GYF domain of CD2BP2, which is also flexible and not visualized in their structure. The authors could also map CD2BP2's D2 domain at the lower part of the PRP8 N domain, but its possible function, based on its location, remains unclear. In summary, the cryo-EM structure of the CD2BP2-containing 20S U5 snRNP is well done. However, we do not learn much about the role of CD2BP2 in the assembly of U5 and the tri-snRNP complexes. Additional structural and/or biochemical experiments along the lines discussed below are clearly needed in order to substantiate the current models that the authors present concerning the function of CD2BP2.

1. The authors discuss a potential role of CD2BP2 in recruiting PRP6 to the U5 snRNP particle, but unfortunately, the 20S U5 structure does not provide conclusive hints as to its mechanism. Extended Data Figure 8d shows a cartoon of an AlphaFold2 model of the CD2BP2-PRP6 binary complex. The authors propose a putative location for such a CD2BP2-PRP6-TPR complex in a low-resolution EM density map, but the latter does not appear to be large enough to fit the entire PRP6-TPR domain. The authors should perform protein cross-linking coupled with mass spectrometry, which could be helpful to better characterize the location of proteins in the poorly resolved densities, as well as the possible position of DIM1 in the complex.

2. The authors should investigate more thoroughly the protein composition of the 20S U5 snRNP which they purify using the CD2BP2 knockout cell line. It could be potentially very interesting to see whether PRP6 (and DIM1) are bound to U5 snRNP in the absence of CD2BP2.

3. The authors mention that Aar2 accumulates in the snRNP preparation they obtain from extracts of the CD2BP2 knockout cell line. It would also be interesting to learn more about the possible interplay between CD2BP2 and AAR2 during the assembly of the U5 snRNP particle.

Additional points:

4. The authors propose that the displacement of CD2BP2 is the result of a large-scale movement of the PRP6-TPRs upon U4/U6 di-snRNP recruitment. However, it is not shown how and why the stabilization of the PRP6-TPR on the tri-snRNP would displace CD2BP2 or liberate DIM1.

5. The FSC plot from Extended data Fig. 4d does not match the proposed resolutions in Extended data Fig. 3. From Extended data Fig. 4d, state-1 looks like 3.6-3.8 Å, state-2 20 Å, and state-3 4.8 Å. Map-to-model FSC also suggests the state-1 map resolution is somewhere close to 3.8 Å.

6. Additional 3D classification could be performed on state-1 to further dissect the various positions of BRR2 in this complex.

7. Extended data Fig. 4 has two "d" panels.

Author Rebuttal to Initial comments

First of all, we would like to thank both reviewers for their insightful comments and suggestions on how to improve this manuscript. Our point-to-point answers to the specific comments are embedded in the text below.

Reviewer #2

The paper is concise and well written. The figures are clear. Overall, this work provides new insights into the role of CD2BP2 in 20S U5 snRNP biogenesis, which will be of general interest for the splicing field and the RNA community. Publication of this work in NSMB is supported, provided that the following points are addressed:

We appreciate this supportive comment

Major points

1. After inspection of the provided atomic coordinates and cryo-EM maps, a couple of points should be addressed to improve the quality of the published structural data:

- Any ligand without defined cryo-EM density to support their presence should be removed (e.g. IHP).

IHP has been removed now.

- Side chains in low quality or low-resolution regions should be fully trimmed to the C-beta position (BRR2, DDX23, Sm proteins, etc).

Side chains were trimmed accordingly.

- The atomic model for the "complete" 20S U5 snRNP seems to be a composite of the refined "core" atomic model and additional rigid body fitted proteins (BRR2, DDX23, etc). This is okay but B-factors (ADP) must be refined in the local-filtered map so that they reflect the local resolution from the EM data.

This has been modified accordingly.

- In the "core" atomic model, chain D Arg 73 is floating in space.

This has been corrected now.

2. There are inconsistencies between the provided coordinates, refinement statistics and FSC curves presented in the Extended Data figures:

- Extended Data Fig. 4: panel a shows the FSC curve of "3.1Å" (which is "State I" based on Extended Data Fig. 3 and should be indicated as such in the legend) while panel d presents FSC curves for 3 determined structures. FSC curves for State I in both panels a and d don't match.

We agree that these data have not been presented very clearly, and we thank the reviewer for pointing out the mismatch between the FSC curve in Extended Data Fig. 4a and d. Panel d was plotted manually, and by mistake the x-axis values were taken from the wrong column, resulting in the observed discrepancy. We have now labelled plots more clearly and corrected the mismatch.

- Extended Data Figure 5: panel b, model-to-map FSC should be plotted to at least Nyquist frequency.

This has been corrected now. The new calculation has been done within Phenix with the coordinates modified following reviewers' instructions.

- Table 2: the refinement statistics and reported refinement software (Refmac) do not match the header of the provided PDB files (Phenix). The indicated Clashescore is also relatively high for this kind of resolution and does not match the PDB header. Overall, the authors should recheck the provided processing and refinements data for consistency.

We thank the reviewer for spotting this discrepancy. Table 2 was prepared from the initial coordinates, which were subsequently improved by changing the refinement protocol, but the statistics were not updated due to our omission. This has been corrected now, and improved statistics are presented in Table 2.

3. The authors should provide a description of the Western blotting and In-vitro splicing assay protocols in the Methods section.

This has been added.

Minor points

1. The authors mention TSSC4 being part of the immature/recycled 20S U5 snRNP. Additionally, this factor is present in the purified 20S U5 snRNP (mass spectrometry analysis). However, TSSC4 was not identified nor visualized in the cryo-EM maps. This is discussed in the legend of Extended Fig. 6 but this is actually an important point and could be worth including in the main text if space permits.

We agree with the reviewer that this point is an interesting point that could be better highlighted. We included a brief note that TSSC4 is not visible in the structure when the TSSC4 is introduced:

"The composition of the complex is in good agreement with previous reports^{7,17} and interestingly includes an additional assembly factor, TSSC4 (not resolved in the structure)^{8,9}."

We also included an additional reference to ED Fig. 6, when the potential earlier intermediate (state II) is discussed. To avoid breaking the flow of the paper we kept the detailed discussion about TSSC4 in the Extended Data.

2. Extended Data Fig. 8: panel d, the two protein chains should be colored based on the AlphaFold pLDDT score so that the reader can appreciate the confidence of the predicted protein interfaces.

An additional panel has been added to this figure, as suggested

6. Typo: top of page 2, "Extended Data Fig. 87".

This has been corrected now.

Reviewer #3:

Remarks to the Author:

The U4/U6.U5 tri-snRNP is the largest building block of the spliceosome. It assembles from a 20S U5 snRNP and a U4/U6 di-snRNP particle. A hallmark of the 20 S U5 snRNP is the protein CD2BP2 that is believed to play a role in the biogenesis of the tri-snRNP. While a cryo-EM structure of the human tri-snRNP has been available for several years, a high-resolution structure of the 20S U5 snRNP has been missing. In this manuscript, Galej and coworkers succeeded in reconstructing a 3.1 Å cryo-EM structure of a 20 S U5 snRNP particle, which was purified using a tagged CD2BP2 protein. The key observation in this manuscript is that CD2BP2's N-terminal D1 domain straddles the PRP8 Large and N domains and thus stabilizes the orientation of these two PRP8 domains towards each other. Moreover, the binding of CD2BP2-D1 to PRP8 is mutually exclusive with the binding of the U5 protein DIM1, nicely explaining why the authors could not map DIM1 in their structure. As DIM1 is well represented in the MS analysis of the purified U5 snRNP, the authors assume that DIM1 may be tethered to the C-terminal GYF domain of CD2BP2, which is also flexible and not visualized in their structure. The authors could also map CD2BP2's D2 domain at the lower part of the PRP8 N domain, but its possible function, based on its location, remains unclear. In summary, the cryo-EM structure of the CD2BP2-containing 20S U5 snRNP is well done. However, we do not learn much about the role of CD2BP2 in the assembly of U5 and the tri-snRNP complexes. Additional structural and/or biochemical experiments along the lines discussed below are clearly needed in order to substantiate the current models that the authors present concerning the function of CD2BP2.

We would like to thank the reviewer for the critical assessment of our manuscript.

1. The authors discuss a potential role of CD2BP2 in recruiting PRP6 to the U5 snRNP particle, but unfortunately, the 20S U5 structure does not provide conclusive hints as to its mechanism. Extended Data Figure 8d shows a cartoon of an AlphaFold2 model of the CD2BP2-PRP6 binary complex. The authors propose a putative location for such a CD2BP2-PRP6-TPR complex in a low-resolution EM density map, but the latter does not appear to be large enough to fit the entire PRP6-TPR domain. The authors should perform protein cross-linking coupled with mass spectrometry, which could be helpful to better characterize the location of proteins in the poorly resolved densities, as well as the possible position of DIM1 in the complex.

As suggested, we performed a BS3 cross-linking mass spectrometry experiment to address this issue. The data is now included in the Extended Data Figure 9. The cross-linking protocol

was validated by the analysis of Ca-Ca distances for the cross-linked residues visible in the core structure. Most of the high-confidence cross-links (XLs) fall within the typically used 35 Å cut-off distance, satisfying the chemical nature of the BS3 and accounting for some protein dynamics. There are few cross-links violating this distance, but they are located in proteins, which are highly dynamic in our structure (i.e. DDX23) and likely represent alternative conformational states.

As for the location of proteins poorly resolved in the map, we analysed cross-links between CD2BP2, PRP6, DIM1 and 40K from two independent experiments. As expected, we could detect XLs between DIM1 and GYF domain of CD2BP2 (consistent with their crystal structure), and DIM1 and N-terminus of PRP6 (consistent with the arrangement observed in the tri-snRNP). We detect multiple cross-links between PRP6^{TPR} domain and CD2BP2, consistent with their interaction proposed here and by previous studies.

Interestingly C-terminus of CD2BP2 shows multiple cross-links to 40K, which suggests their close proximity. This would be in agreement with its putative location in the unassigned density of our reconstruction (Extended Data Figure 8). We agree with the reviewer that the unassigned density mentioned here is not large enough to fit both PRP6 and CD2BP2^{GYF}-DIM1, that's why we left the interpretation ambiguous indicating that it belongs to one and/or part of both. Based on additional data we conclude now that the extra density most likely belongs to CD2BP2^{GYF} bound to DIM1. We have rephrased it accordingly to address this comment. Noteworthy, regardless of the exact assignment of this weak density, cross-linking data still indicates that PRP6^{TPR} is located in close proximity to the CD2BP2. Therefore, our interpretation concerning its sequestration and relocation upon U4/U6 snRNP binding remains valid.

2. The authors should investigate more thoroughly the protein composition of the 20S U5 snRNP which they purify using the CD2BP2 knockout cell line. It could be potentially very interesting to see whether PRP6 (and DIM1) are bound to U5 snRNP in the absence of CD2BP2.

We agree with the reviewer that the composition of this complex is worth investigating. However, the difficulty in studying it arises from the fact that CD2BP2 is a hallmark protein of the 20S U5 snRNP, and in its absence, specific purification of the 20S U5 snRNP is not possible. This is the primary reason why we analysed the KO effect in the TMG pull-down fractions containing all snRNPs. Indeed, as suggested by the reviewer, PRP6 is depleted in the KO condition (Fig. 1A, top hit of the "U5 proteins"). However, as we discussed in the manuscript, the effect is only moderate, and generally most of the other U5 snRNP and tri-snRNP proteins are also depleted in the snRNPs purified from CD2BP2^{KO} cell line, therefore the effect is not specific to PRP6. DIM1 could not be detected therefore we could not comment on its changes.

To address the reviewer's comment, we investigated this problem further using two approaches:

1. Western blotting of the glycerol gradient fractions of nuclear extracts from WT and CD2BP2^{KO} cell lines. Size fractionation allowed us to identify fractions containing 20S U5 snRNP. As expected, DIM1 is significantly depleted in those fractions. We could not see any clear differences for the PRP6 recruitment, suggesting that contacts of PRP6 N-terminus with PRP8 and U5 snRNA (as observed in the 20S U5 snRNP and

tri-snRNP structures) might be sufficient to tether it to the complex. This analysis has been now included in Extended Data Figure 1e.

2. *We hypothesised that some of the CD2BP2 knock-out effects specific to PRP6 and DIM1 might be masked during the long-term depletion (KO) due to the redundancy of the assembly pathways and/or compensatory effects. Therefore, we attempted to analyse the composition of snRNPs assembled de novo in the WT or KO cell lines using transient (24h) expression of a GFP-tagged SNU114 (SNRNP116), a core component of U5 snRNP. We performed GFP nanobody pull-downs in triplicates and analysed SNU114-bound proteins using quantitative proteomics. While KO cell line expressed SNU114 even better than the WT, snRNPs assembly was largely unaffected in the absence of CD2BP2. Most of the U5 and tri-snRNP components are consistently recruited more efficiently to the SNU114 in the wt cell line, the differences are very small. This result echoes the TMG-pull down experiment presented in Figure 1. Since there is no new information in this experiment, we decided not to include it in the manuscript to avoid redundancy.*

Revision Figure 1. Quantitative proteomics of *de novo* assembled snRNPs purified via transient expression of the GFP-tagged SNU114 (SNRNP116). **a)** GFP-tagged SNU114 was transiently expressed in WT and CD2BP2-KO HEK293T cell lines. **b)** Samples for the mass spectrometry analysis were purified using GFP nanobodies in triplicates and subjected to the TMT-plex MS workflow.

3. The authors mention that Aar2 accumulates in the snRNP preparation they obtain from extracts of the CD2BP2 knockout cell line. It would also be interesting to learn more about the possible interplay between CD2BP2 and AAR2 during the assembly of the U5 snRNP particle.

We agree with the reviewer that the interplay between AAR2 and CD2BP2 is an important point of the paper. While in the original version of the manuscript, we observed that AAR2 is enriched in snRNPs purified from CD2BP2^{KO} cell line, we could not differentiate whether it is due to its upregulation or inability to be displaced in the absence of CD2BP2. Measuring AAR2 levels in nuclear extract proved to be challenging due to a lack of reliable and specific antibodies (several tested) therefore we addressed this issue by performing quantitative mass spectrometry on the complete nuclear extracts from the WT and CD2BP2-KO cell lines. We observe only minor differences in the AAR2 levels between the WT and KO conditions. In contrast, the previously reported differences are much larger in the TMG-purified fractions. This suggests that previously observed differences more likely originate in AAR2 being stuck on the U5 snRNP precursor in the absence of CD2BP2 rather than its upregulation. This analysis has been now included in Extended Data Figure 1f.

To follow up on that, we attempted to displace AAR2 from the precursor U5 snRNP particle by the addition of recombinant CD2BP2. We established a stable cell line expressing GFP-tagged AAR2, which was used to purify precursor U5 snRNP complexes. As previously reported (Malinova et al., 2017, Coulter et al., 2017) AAR2 indeed co-purifies PRP8, BRR2, SNU114 and small amounts of the U5 snRNA. We immobilised these complexes on the GFP nanobody resin and treated them with purified CD2BP2 (Figure 2). Under the conditions used in this experiment, we could not observe any release of the U5 snRNP proteins from the GFP-resin. This suggests that CD2BP2 cannot readily exchange AAR2 in vitro. It is possible that other protein and/or posttranslational modifications (as previously suggested, Boon et al., NSMB 2007) might be needed to achieve this process.

Revision Figure 2. In vitro displacement of AAR2 from PRP8. a) AAR2-containing U5 snRNP complex purified from HEK293F cells. b) experimental workflow for the AAR2-CD2BP2 exchange experiment, c) SDS-PAGE analysis of the experiment from (b). Lanes 1 and 2, are the input samples AAR2-U5 and recombinantly expressed CD2BP2. * denotes small amounts of PRP8 and BRR2 that are still associated with CD2BP2 purified under stringent high-salt conditions, but the majority of this sample consists of free CD2BP2. In the case of successful AAR2 displacement by CD2BP2, one would expect the release of PRP8 and BRR2 from the GFP resin. Under the conditions tested, we could not detect any displacement of these two proteins, and they appear to stay associated with the GFP resin independently of the amount of CD2BP2 used.

To gain more insights into AAR2 function, we also attempted CRISPR/Cas9 KO of the AAR2 in the wt and CD2BP2^{KO} cell lines. In both cases, we could not recover any positive clones, suggesting that such a deletion might be lethal for cells.

While the question of the exact interplay between AAR2 and CD2BP2 remains open, we believe it goes beyond the scope of our brief communication.

Additional points:

4. The authors propose that the displacement of CD2BP2 is the result of a large-scale movement of the PRP6-TPRs upon U4/U6 di-snRNP recruitment. However, it is not shown how and why the stabilization of the PRP6-TPR on the tri-snRNP would displace CD2BP2 or liberate DIM1.

We now rephrase this section to make it clearer:

“Since PRP6 and CD2BP2 interact with one another (Extended Data Fig. 8 and 9), such movement of PRP6 could exert a force on CD2BP2, displacing it from PRP8 and liberating DIM1 allowing it to adopt its final location. “

5. The FSC plot from Extended data Fig. 4d does not match the proposed resolutions in Extended data Fig. 3. From Extended data Fig. 4d, state-1 looks like 3.6-3.8 Å, state-2 20 Å, and state-3 4.8 Å. Map-to-model FSC also suggests the state-1 map resolution is somewhere close to 3.8 Å.

We apologise for this mistake. Panel 4d was plotted manually, and by mistake the x-axis values were taken from the wrong column, resulting in the observed discrepancy, also pointed out by reviewer #2. We have now labelled plots more clearly and corrected this mistake.

6. Additional 3D classification could be performed on state-1 to further dissect the various positions of BRR2 in this complex.

We attempted to perform an additional classification of this region as suggested. The signal for BRR2 is very weak, and we could not separate any meaningful differences and/or stable sub-states from the particles present in this class. The results of this classification are summarised in the figure below. Since we could not separate any meaningful classes, we decided to not include this data in the final manuscript.

Revision Figure 3. Additional 3D classification of the main 20S U5 snRNP state (state I). Classification without image alignment was performed with the mask focused on the BRR2 region.

7. Extended data Fig. 4 has two “d” panels.

This has been corrected

Decision Letter, first revision:

Message: Our ref: NSMB-BC48096A

20th Dec 2023

Dear Dr. Galej,

Thank you for submitting your revised manuscript "Structure of the human 20S U5 snRNP" (NSMB-BC48096A). It has now been seen by the original referees and their comments are below. Both reviewers have let us know that they think that the paper has further improved in revision and is ready for publication, therefore we'll be happy to accept it in principle in Nature Structural & Molecular Biology, pending minor revisions to satisfy the final request of reviewer 2 and to comply with our editorial and formatting guidelines.

We are now performing detailed checks on your paper and will send you a checklist detailing our editorial and formatting requirements at the beginning of January. Please do not upload the final materials and make any revisions until you receive this additional information from us.

To facilitate our work at this stage, it is important that we have a copy of the main text as a word file. If you could please send along a word version of this file as soon as possible, we would greatly appreciate it; please make sure to copy the NSMB account (cc'ed above).

Sincerely,

Dimitris Typas
Associate Editor
Nature Structural & Molecular Biology
ORCID: 0000-0002-8737-1319

Reviewer #2 (Remarks to the Author):

The revised version of the manuscript entitled "Structure of the human 20S U5 snRNP" by Schneider et al. has addressed all the concerns raised during the previous review. The revised atomic models, together with the amended extended figures, methods, and tables, have improved the quality and readability of the manuscript. However, the authors should be more cautious about their interpretation of the CD2BP2-PRP6 interaction predicted by AlphaFold multimer. Based on the newly provided pLDDT score in ED Fig. 8e, the model confidence is very low for the second interaction highlighted by the authors in panel d (right box). As such, this very low confidence interaction should be removed from ED Fig.8 in panels a and d. With this minor point addressed, this work is now suitable for publication.

Author Rebuttal, first revision:

Following reviewer's suggestion we modified ED Fig8. by removing one of the CD2BP2-PRP6 predicted binding site from panels a and d.

Final Decision Letter:

Message: 14th Feb 2024

Dear Dr. Galej,

We are now happy to accept your revised paper "Structure of the human 20S U5 snRNP" for publication as a Brief Communication in Nature Structural & Molecular Biology.

Note the policy of the journal on data deposition:

<http://www.nature.com/authors/policies/availability.html>.

Your paper will be published online soon after we receive proof corrections and will appear in print in the next available issue. You can find out your date of online publication by contacting the production team shortly after sending your proof corrections.

Please note that *Nature Structural & Molecular Biology* is a Transformative Journal (TJ). Authors may publish their research with us through the traditional subscription access route or make their paper immediately open access through payment of an article-processing charge (APC). Authors will not be required to make a final decision about access to their article until it has been accepted. Find out more about Transformative Journals

Sincerely,

Dimitris Typas
Associate Editor
Nature Structural & Molecular Biology
ORCID: 0000-0002-8737-1319